
# Numerical integrators for Lagrangian oceanography

Tor Nordam[1,2] and Rodrigo Duran[3,4]

[1]SINTEF Ocean, Trondheim, Norway
[2]Department of Physics, Norwegian University of Science and Technology, Trondheim, Norway
[3]National Energy Technology Laboratory, Albany, OR 97321, USA
[4]Theiss Research, San Diego, CA 92037, USA

**Correspondence:** Tor Nordam (tor.nordam@sintef.no)

**Abstract.** A common task in Lagrangian oceanography is to calculate a large number of drifter trajectories from a velocity field pre-calculated with an ocean model. Mathematically, this is simply numerical integration of an Ordinary Differential Equation (ODE), for which a wide range of different methods exist. However, the discrete nature of the modelled ocean currents requires interpolation of the velocity field, and the choice of interpolation scheme has implications for the accuracy and efficiency of

the different numerical ODE methods.

We investigate trajectory calculation in modelled ocean currents with $800\,\mathrm{m}$, $4\,\mathrm{km}$, and $20\,\mathrm{km}$ horizontal resolution, in combination with linear, cubic and quintic spline interpolation. We use fixed-step Runge-Kutta integrators of orders 1–4, as well as three variable-step Runge-Kutta methods (Bogacki-Shampine 3(2), Dormand-Prince 5(4) and 8(7)). Additionally, we design and test modified special-purpose variants of the three variable-step integrators, that are better able to handle discontinuous

derivatives in an interpolated velocity field.

Our results show that the optimal choice of ODE integrator depends on the resolution of the ocean model, the degree of interpolation, and the desired accuracy. For cubic interpolation, the commonly used Dormand-Prince 5(4) is rarely the most efficient choice. We find that in many cases, our special-purpose integrators can improve accuracy by many orders of magnitude over their standard counterparts, with no increase in computational effort. The best results are seen for coarser

resolutions ($4\,\mathrm{km}$ and $20\,\mathrm{km}$), thus the special-purpose integrators are particularly advantageous for research using regional to global ocean models to compute large numbers of trajectories. Our results are also applicable to trajectory computations from atmospheric models.

## 1  Introduction

Calculating trajectories of tracers through a pre-calculated velocity field is a common task for many applications (van Sebille et al., 2018). Oceanic and atmospheric transport simulations are frequently built on this approach, and used to calculate, for example, the transport of pollutants (see, e.g., Rye et al. (1998); North et al. (2011); Povinec et al. (2013); Onink et al. (2019)),





distribution of algae and plankton (see, e.g., Siegel et al. (2003); Woods (2005); Visser (2008)), search and rescue operations (see, e.g., Breivik and Allen (2008); Serra et al. (2019)), or temperature and salinity pathways (see, e.g., Barkan et al. (2017)).

Similarly, climate change studies may compute vast numbers of trajectories to understand transport of heat and salt (see, e.g., Dugstad et al. (2019)). Computation of trajectories for a variety of atmospheric species are also a common application (see, e.g., Sirois and Bottenheim (1995); Riuttanen et al. (2013); Nieto and Gimeno (2019)). Other applications include the calculation of Lagrangian Coherent Structures (LCS), which is not a transport simulation *per se*, but which still uses tracer trajectories to analyze flow fields (see, e.g., Farazmand and Haller (2012); Onu et al. (2015); Haller (2015); Duran et al. (2018)).

For all of these applications, it is of interest to obtain trajectories of the desired accuracy with minimal computational work, or conversely, to obtain the most accurate solution possible for a given amount of computational effort. Marine and atmospheric transport applications often require computing large numbers of trajectories, which are essentially solutions of an ordinary differential equation (ODE). As this can be computationally quite demanding, guidance on how to select the optimal combination of numerical schemes for a given application is of practical value.

We further note that in ODE parlance, the velocity fields represented by ocean currents (and wind) may be both stable and unstable, often presenting hyperbolic points capable of exponentially growing initially small errors. It is therefore recommended to use higher-order integration methods, or small time steps with lower-order integration methods. This is particularly relevant for long integration times (months to years) where error accumulates and can be amplified.

In the applied mathematics community, a standard first choice for numerically solving an ODE is a variable-step integrator
(see, e.g., Gladwell et al. (2003)). Variable-step integrators use clever choices of function evaluations in order to evaluate the local error in each step of the solution, and the time-step is dynamically chosen to be as long as possible while meeting a prescribed error estimate. Thus, variable-step integrators tend to be more efficient than their fixed-step counterparts.

However, there is limited discussion of such an approach in the literature on applied Lagrangian oceanography. Integrators used in marine transport applications may range from Euler's method (see, e.g., Zelenke et al. (2012); De Dominicis et al.
(2013)), to a more typical 4th-order Runge-Kutta method (see, e.g., García-Martínez and Flores-Tovar (1999)). Some alternatives seek to cut on computational time by using less evaluations, like the 4th-order Milne-predictor, Hamming-corrector integration scheme (see, e.g., Narváez et al. (2012)), or the 4th-order Adams-Bashforth method (see, e.g., Yang et al. (2008)).

In the context of LCS, variable-timestep integrators appear to be a more common, yet not universal, choice. Interpolation schemes, which must be used to evaluate discretely gridded velocity fields at arbitrary points, have also received some attention
in the LCS field. Ali and Shah (2007) use a 4th-order Runge-Kutta-Fehlbergh method and the local cubic interpolation recipe of Lekien and Marsden (2005). Beron-Vera et al. (2008) use linear interpolation and the classic 4th-order Runge-Kutta. Shadden and Taylor (2008) use linear basis functions for interpolation, and a Runge-Kutta-Fehlberg scheme for integration. Peng and Dabiri (2009) use the 4th-order Runge-Kutta with a velocity field derived from Particle Image Velocimetry (PIV), though with no interpolation scheme specified.

Solving diverse types of marine-transport problems is very common, and given the vast number of computations that are often involved, it seems natural to ask how variable-step integrators perform. Because a pre-calculated velocity field is necessarily given at discrete times and spatial locations, interpolation must be used to create continuous representations of these





velocity fields that can then be integrated using numerical schemes. In practice, the choice of an interpolation scheme will have implications for the accuracy that can be achieved with the different numerical integrators, as well as the computational effort.

In this paper, we compare several approaches for interpolation of the velocity field, and numerical integration of the trajectories. We include both fixed and variable stepsize integrators. As input data to the trajectory calculations, we use modelled ocean currents at $20\,\mathrm{km}$, $4\,\mathrm{km}$ and $800\,\mathrm{m}$ resolutions. These are representative of current high-resolution Earth Modeling Systems, regional (eddy-resolving) ocean models and submesoscale-resolving ocean models, respectively (Lévy et al., 2012), and thus span a wide range of applications.

The layout of this paper is as follows: In Section 2, we introduce some theory on numerical integration of ODEs, including a description of the interpolation and integration schemes used, and a discussion of the local and global error of numerical integrators. Next, in Section 3 we discuss the performance of numerical integrators for velocity fields with discontinuous derivatives, and describe how we modified well-known variable-step integrators to improve their performance for this particular application. Section 4 describes how the interpolation and integration schemes were implemented in code, and the numerical

experiments that were carried out. Section 5 contains the results of our investigation, and a discussion of the results, and finally in Section 6 we present some conclusions on the most efficient choice of integrator for different applications.

## 2   Theory

The topic of the current paper is to study the numerical calculation of tracer advection by precalculated, gridded velocity fields, with a focus on applications in Lagrangian oceanography. Note that we ignore diffusion, and consider pure advection with

ocean currents. In this case, the trajectory of a particle being advected passively through a velocity field is defined by the ODE

$$\dot{\mathbf{x}} = \mathbf{v}(\mathbf{x}, t), \tag{1}$$

where $\mathbf{v}(\mathbf{x}, t)$ is the velocity at position and time $(\mathbf{x}, t)$, along with an initial condition, $\mathbf{x}(t_0) = \mathbf{x}_0$. Such a problem is called an initial value problem, and solving it means to find the value of $\mathbf{x}(t)$ at later times, $t > t_0$.

Finding the solution of an initial value problem by numerical means is known as numerical integration of the differential

equation. A large body of literature exists on the topic of numerical integration, and a range of different techniques exist, both general-purpose methods that work with many different problems (see, e.g., Hairer et al. (1993); Hairer and Wanner (1996)), and special-purpose methods that for example preserve some symmetry of the problem (see, e.g., Hairer et al. (2006)). In this paper, we will consider both fixed- and variable-step methods from the Runge-Kutta family.

Common to all numerical ODE methods is that they make discrete steps in time. In a fixed-step method, time is incremented

by a fixed amount, $h$, at each iteration, and we have

$$t_n = t_0 + nh. \tag{2}$$





For the variable-step methods, the value of the timestep may change throughout the simulation, such that $t_n + h_n = t_{n+1}$. Hence, the relationship between time and timestep in this case becomes

$$t_n = t_0 + \sum_{i=0}^{n-1} h_i, \ \ n \geq 1. \tag{3}$$

For both types of methods, if the solution is to be calculated up to time $t_N$, we adjust the last timestep as necessary to, stop the integration exactly at $t_N$:

$$h_{N-1} \to \min(h_{N-1}, t_N - t_{N-1}). \tag{4}$$

Finally, we will use notation where we let $x_n$ denote the numerically obtained solution at time $t_n$, and we let $x(t_n)$ be the *true* solution at time $t_n$. Note that while $x(t_n)$ is usually not known, we will still assume that there exists a unique, true solution

(Hairer et al., 1993, pp 35–43).

## 2.1   Error Bounds

Since numerical integration is most commonly used in situations where the exact solution is unknown, it becomes necessary to estimate the error by purely numerical means. In general, the idea is that a smaller timestep, $h$, gives a more accurate solution, and as $h \to 0$, the numerically obtained solution converges to the true solution. The rate of convergence depends on the chosen

integration method.

There are two important measures of the error: The *local* error and the *global* error. The local error is the error made in a single step. Assume there is no error in the position at time $t_{n-1}$, that is, $x(t_{n-1}) = x_{n-1}$. Then, the local error in step $n$ is given by (Hairer et al., 1993, p. 156)

$$e(h) = x(t_n) - x_n. \tag{5}$$

The global error, on the other hand, is the error at the end of the computation, at time $t_N$ (assuming $x(t_0) = x_0$), and is given by (Hairer et al., 1993, p. 159)

$$E(h) = x(t_N) - x_N. \tag{6}$$

It can be shown that for a Runge-Kutta method of order $p$, and for an ODE given by $\dot{x} = f(x,t)$, where all partial derivatives of $f(x,t)$ up to order $p$ exist and are continuous (that is, $f \in C^p$), the local error is bounded by

$$|x(t_0 + h) - x_1| \leq Ch^{p+1}, \tag{7}$$

where $C$ is some constant, which depends on the method and on the partial derivatives of $f(x,t)$ (Hairer et al., 1993, p. 157). If the local error is $\mathcal{O}(h^{p+1})$, then the global error will be $\mathcal{O}(h^p)$ (Hairer et al., 1993, pp. 160–162). When the global error is proportional to $h^p$, the method is said to be of order $p$.





## 2.2 Numerical integration methods

We have chosen to consider seven different numerical integration schemes, all from the family of Runge-Kutta methods. These include four methods with fixed timestep:

- 1st-order Runge-Kutta (Euler's method),
- 2nd-order Runge-Kutta (explicit trapezoid),
- 3rd-order Runge-Kutta (Kutta's method),
- 4th-order Runge-Kutta,

For details of these methods, we refer to, e.g., Griffiths and Higham (2010, pp. 24, 44–45, and 131). We have also considered three methods with variable timestep:

- Bogacki-Shampine 3(2),
- Dormand-Prince 5(4),
- Dormand-Prince 8(7).

For further details of these methods, we refer to Bogacki and Shampine (1989), and Dormand and Prince (1980, 1986).

As an example, and to aid the explanation of the timestep adjustment routine which will follow in Sections 2.3 and 3.3, we will describe the Bogacki-Shampine 3(2) method in some detail. For an ODE given by

$$\dot{x} = f(x,t), \tag{8}$$

the Bogacki-Shampine 3(2) method, for making a step from position $x_n$, at time $t_n$, to position $x_{n+1}$, at time $t_{n+1} = t_n + h_n$, is

$$k_1 = f(x_n, t_n) \tag{9a}$$
$$k_2 = f(x_n + \frac{1}{2}k_1 h_n, t + \frac{1}{2}h_n) \tag{9b}$$
$$k_3 = f(x_n + \frac{3}{4}k_2 h_n, t + \frac{3}{4}h_n) \tag{9c}$$
$$k_4 = f(x_n + \frac{2}{9}k_1 h_n + \frac{1}{3}k_2 h_n + \frac{4}{9}k_3 h_n, t + h_n) \tag{9d}$$
$$\hat{x}_{n+1} = x_n + \frac{7}{24}k_1 h_n + \frac{1}{4}k_2 h_n + \frac{1}{3}k_3 h_n + \frac{1}{8}k_4 h_n \tag{9e}$$
$$x_{n+1} = x_n + \frac{2}{9}k_1 h_n + \frac{1}{3}k_2 h_n + \frac{4}{9}k_3 h_n. \tag{9f}$$

This provides two estimates of the next position, of which $x_{n+1}$ is of order 3, and $\hat{x}_{n+1}$ is of order 2. For this method, the higher order estimate is used to continue the integration (known as local extrapolation, see Hairer et al. (1993, p. 168)), while
the lower order estimate is used to calculate $|x_{n+1} - \hat{x}_{n+1}|$, which is used to estimate the local error and adjust the timestep (see Section 2.3).





Comparing Eqs. (9d) and (9f), we note that $k_4 = f(x_{n+1}, t_{n+1})$. Hence, the weights of this method are chosen such that $k_4$ at one step is equal to $k_1$ at the next step. This property is known as First Same As Last (FSAL), and saves one evaluation of the right-hand side for every step after the first (see, e.g., Hairer et al. (1993, p. 167)). Hence, with only three new evaluations of $f(x, t)$, this method can provide both a third-order estimate used to continue the integration, and a second order estimate for error control.

Dormand-Prince 5(4) uses 7 evaluations of $f(x, t)$, to construct a 5th-order estimate for continuing the integration, and a 4th-order estimate for error control and timestep adjustment. This method also has the FSAL property, meaning that it uses only 6 evaluations for every step after the first. The final method considered, Dormand-Prince 8(7), uses 13 evaluations of $f(x, t)$ to construct 8th-order and 7th-order estimates, of which the 8th-order is used to continue the integration. This integrator does not have the FSAL property.

## 2.3 Timestep adjustment

In the code used to carry out numerical experiments, timestep adjustment has been implemented based on the description in Hairer et al. (1993, pp. 167–168). The user must specify two tolerance parameters, the absolute tolerance, $T_A$, and the relative tolerance, $T_R$. We then want the estimate of the local error to satisfy

$$|x_{n+1} - \hat{x}_{n+1}| \le T_A + T_R \cdot \max(x_n, x_{n+1}). \tag{10}$$

To provide a measure of the error, we introduce $\bar{e}$, which is a normalised numerical estimate of the true local error (Eq. (5)), given by

$$\bar{e} = \sqrt{\sum \left( \frac{x_{n+1} - \hat{x}_{n+1}}{T_A + T_R \cdot \max(|x_n|, |x_{n+1}|)} \right)^2}, \tag{11}$$

where in our case we take the sum over the two vector components of the solution. We would like to find the optimal timestep, in the sense of giving the optimal balance between error and computational speed. We consider this to be the timestep where the estimated local error is equal to error allowed by the tolerance, in which case we have $\bar{e} = 1$. If, after calculating $\hat{x}_{n+1}$ and $x_{n+1}$, we find that $\bar{e} \le 1$, the step is *accepted*, we update the time to $t_{n+1} = t_n + h_n$, and proceed with the calculation from the new position $x_{n+1}$. If, on the other hand, $\bar{e} > 1$, the step is *rejected*, and we remain at position $x_n$, and attempt to make the step again with a reduced timestep.

For both accepted and rejected steps, we adjust the timestep after every step. Since $\bar{e}$ scales with $h^{q+1}$, where $q$ is the lower order of the two estimates $\hat{x}_{n+1}$ and $x_{n+1}$, we have that the optimal timestep, $h_{\text{opt}}$, is given by (Hairer et al., 1993, p. 168)

$$h_{\text{opt}} = h_n (1/\bar{e})^{\frac{1}{q+1}}. \tag{12}$$

A rejected step represents wasted computational work. Hence, in order to make it more likely that the next step is accepted, we set the timestep to a value somewhat smaller than $h_{\text{opt}}$, and we also seek to prevent the timestep from increasing too fast:

$$h_{n+1} = \min\left(2.5 \cdot h_n, \, 0.8 \cdot h_{\text{opt}}\right) \tag{13}$$





See Hairer et al. (1993, p. 168) for additional discussion of the choice of the factors $0.8$ and $2.5$. The same timestep adjustment routine has been used for all three variable-timestep methods used in this paper.

### 2.4 Interpolation

Modelled ocean current velocity data used in Lagrangian oceanography are commonly provided as vector components given on regular grids of discrete points, $(x_i, y_j, z_k)$, as well as discrete times $t_n$. In order to calculate the trajectory of a particle that moves in the velocity field defined by these data, we will have to evaluate the vector field at arbitrary locations, and (for variable-step methods) arbitrary times. An important point for our purposes is that the local error of an order $p$ Runge-Kutta method is only bounded by $Ch^{p+1}$ if all partial derivatives up to order $p$ of the velocity field, $\mathbf{v}(\mathbf{x}, t)$ in Eq. (1), exist and are

continuous. This has implications for how we should evaluate the gridded velocity field used in a particle transport simulation. For example, if one uses linear interpolation, the first partial derivatives will be constant inside a cell, but discontinuous at cell boundaries. Hence, even for a first-order method the local error is not guaranteed to be bounded by Eq. (7) when stepping across a cell boundary (either in space or time).

In this study, we have chosen to consider three different interpolation schemes, using the same order of interpolation in both

space and time:

- Second order: Linear interpolation

- Fourth order: Cubic spline interpolation

- Sixth order: Quintic spline interpolation

Note that the order of interpolation is 1 plus the polynomial degree (de Boor, 2001, p. 1).

To aid the later discussion, we will briefly explain spline interpolation in one dimension. The generalisation to higher dimensions is quite natural. Assume that we have a grid of $N$ equidistant points, $x_n \in \{x_1, x_2, \ldots, x_{N-1}, x_N\}$, and the values of some function in those points, $y_n = f(x_n)$. The aim of an interpolation procedure is to allow us to approximate the function $f(x)$ at arbitrary $x$, subject to $x_1 \le x \le x_N$. In the case of linear interpolation, the value of the linearly interpolated function, $F_1(x)$ on the interval $[x_n, x_{n+1}]$ is given by

$$F_1(x) = f(x_n) + \frac{x - x_n}{\Delta x} \cdot (f(x_{n+1}) - f(x_n)), \qquad (14)$$

where $\Delta x = x_{n+1} - x_n$ is the grid spacing. We see that $F_1(x)$ is a continuous function, but its derivative, $F_1'(x)$, is not continuous at the grid points.

A cubic spline interpolation, $F_3(x)$, of the same data points as above will be given on an interval $[x_n, x_{n+1}]$ by a cubic polynomial, e.g.,

$$F_3(x) = w_0 + w_1 \tilde{x} + w_2 \tilde{x}^2 + w_3 \tilde{x}^3, \qquad (15)$$

where $\tilde{x} = x - x_n$, and the weights, $w_0$, $w_1$, $w_2$ and $w_3$ are chosen such that $F_3(x)$, $F_3'(x)$, and $F_3''(x)$ are all continuous at the grid points (see, e.g., Press et al. (2007, pp. 120–124)). By the same token, a fifth-degree spline interpolation gives a piecewise



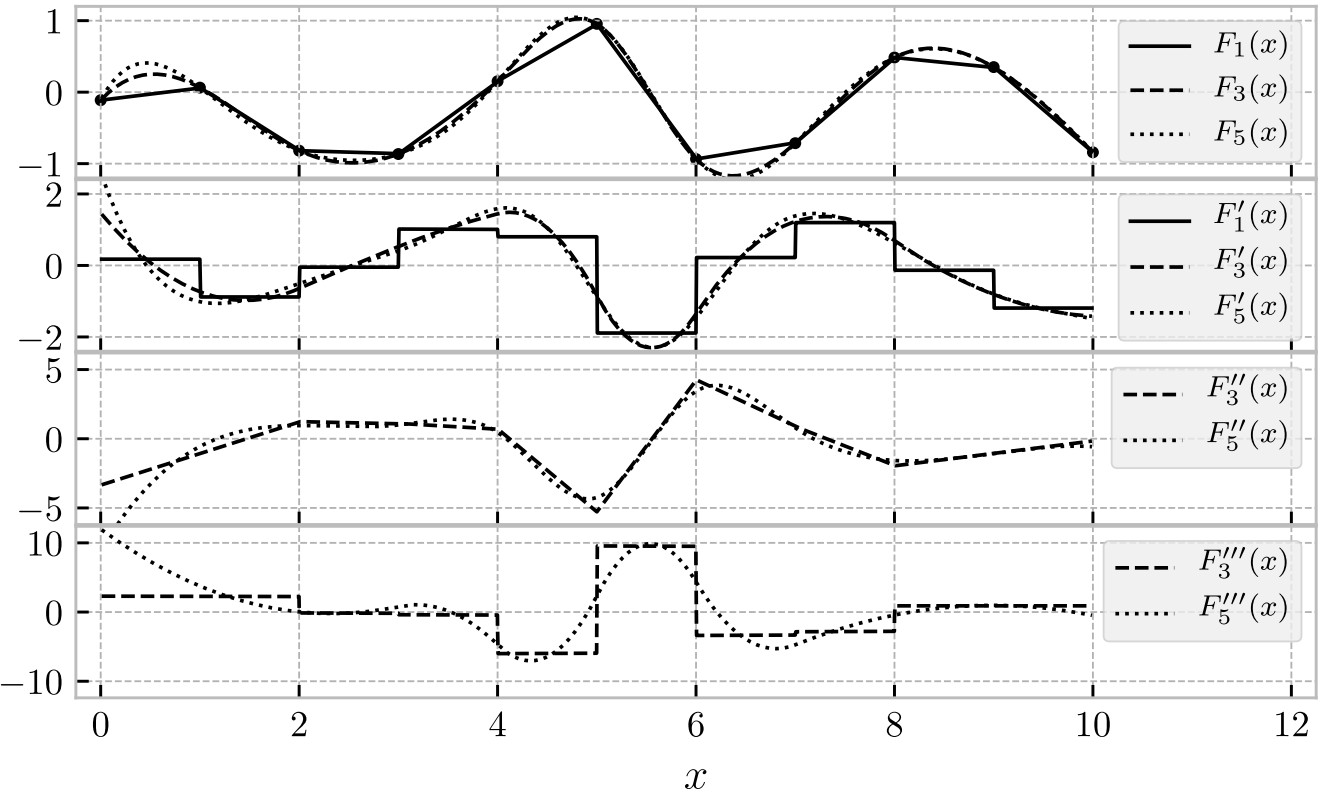

**Figure 1.** One-dimensional illustration of different degrees of interpolation. From the same 11 data points (shown as black circles in the top panel), we have constructed a linear interpolation (continuous lines), a cubic spline interpolation (dashed lines), and a quintic spline interpolation (dotted lines). From the top, the panels show the interpolated functions, the first derivative, the second derivative, and the third derivative. We observe that linear interpolation, $F_1(x)$, gives a discontinuous derivative, and cubic interpolation, $F_3(x)$, gives a discontinuous third derivative.

polynomial function of degree 5, with the property that the first, second, third and fourth derivatives are continuous at the grid points. A one-dimensional illustration of the three degrees of interpolation considered in this paper is provided in Fig. 1. For a

description of how spline interpolation of ocean current velocity fields was implemented, see Section 4.

     Finally, we would like to note two important points on the subject of interpolation in Lagrangian oceanography. First, the purpose of interpolating discrete current data is *not* to approximate the unresolved turbulent motion of the ocean, but simply to provide a consistent recipe for evaluating gridded data at arbitrary locations. Second, once an interpolation scheme has been chosen, one has effectively replaced the gridded input data by a set of analytical expressions, specifying a way in which to

evaluate the velocity field at any point and time. Hence, for a given dataset and interpolation scheme, the initial value problem given by $\dot{\mathbf{x}} = \mathbf{v}(\mathbf{x},t)$, $\mathbf{x}(t_0) = \mathbf{x}_0$, has a unique *true solution* (provided the usual conditions for existence and uniqueness of solutions of ODEs are met, see, e.g., Hairer et al. (1993, pp 35–43)). With increasingly short timestep, $h \to 0$, stable and





consistent numerical integration schemes should converge towards the true solution. However, velocity fields evaluated with *different* orders of interpolation are *not* identical, and will not produce identical trajectories, even as $h \to 0$.

## 3 Special-purpose integrators

In this section, we will discuss the implications of our ODE having a right-hand side with discontinuous derivatives. We consider an analytical example with one discontinuity to illustrate the problem, and present a modified, special-purpose integration routine that handles the discontinuity. We then describe how to implement the same idea in special-purpose variants of regular variable-step integrators, for application in Lagrangian oceanography.

### 3.1 Discontinuous derivatives

As mentioned in Section 2.1, the conditions for a $p$th-order Runge-Kutta method to actually be $p$th-order accurate, require continuous derivatives of the right-hand side, up to and including order $p$. The problem is that consistency of order $p$ of the numerical method is no longer satisfied when the derivatives are not continuous (Kress, 1998, pp. 235, 252). In many cases this means that the error is larger than expected, but in some cases the problem may be more serious: the numerical approximation may be meaningless (Isaacson and Keller, 1994, p. 346).

In practical applications, with interpolated velocity fields, the derivatives are not always continuous. For example, a common choice in the LCS literature appears to be a variable-timestep integrator of order 4 and 5 (see, e.g., Ali and Shah (2007); Shadden et al. (2010); Beron-Vera et al. (2010); Maslo et al. (2020)). Theoretically, seventh-order spline interpolation, yielding five continuous derivatives, are required for the error estimates in the stepsize control routine to hold. However, higher-order spline interpolation is more computationally demanding, and in practice cubic spline interpolation appears to be a common choice. It is also worth noting that, in general, spurious oscillations become increasingly problematic with increasing spline order.

For such cases, there exist strategies to deal with the discontinuities in the right-hand side or (more commonly in our case) its derivatives (Hairer et al., 1987, p. 181). Three possible strategies for dealing with ODEs with discontinuities are outlined by Hairer et al. (1993, pp. 197–198):

I Ignore the discontinuity, and let the variable-stepsize integrator sort out the problem.

II Use an integrator with an error control routine specifically designed to detect and handle discontinuities (see, e.g., Enright et al. (1988); Dieci and Lopez (2012)).

III Use information about the position of the discontinuity to stop and restart integration at that point.

Given that the issue of interpolation and integration is not typically discussed in great detail in applied papers on Lagrangian oceanography, one assumes that most authors implicitly select the first strategy. However, as pointed out by Hairer et al. (1987, p. 181), this is neither the most accurate, nor the most numerically efficient, approach.





## 3.2 Analytical example

To illustrate the effect of discontinuities in the derivative of the right-hand side, we consider the following ODE:

$$\dot{x} = |\sin(\pi t)|, \quad x(t = 0) = 0. \tag{16}$$

In this case, the right-hand side itself is continuous, but its derivative is discontinuous at $t = 1$. This equation has the analytical solution

$$x(t) = \int\limits_0^t |\sin(\pi s)| \, \mathrm{d}s, \tag{17}$$

and if we consider as an example the solution at time $t_N = 2$, we find $x(t_N) = 4/\pi$. Since the exact solution is known, we can find the error in our numerical solutions by using the exact result as a reference. Hence, we can investigate the convergence of our numerical integration scheme, by considering the error as a function of the timestep, $h$.

In Fig. 2, we show the global error in the solution as a function of timestep, $h$, for the 4th-order Runge-Kutta integrator (continuous black line). The error has been calculated for 161 logarithmically spaced timesteps from 1 to $10^{-4}$. Of these 161 timesteps, only 1, $10^{-1}$, $10^{-2}$, $10^{-3}$, and $10^{-4}$ will evenly divide an interval of length 1. This is significant, as we observe that the error scales approximately as $h^4$ for the timesteps 1, $10^{-1}$, $10^{-2}$, and $10^{-3}$, while for the other timesteps it follows a slower $h^2$ scaling. (Note that at $h = 10^{-4}$, the error is dominated by roundoff error (see Appendix A1), which adds up to about $10^{-13}$ after $20\,000$ steps (Press et al., 2007, p. 10).)

The reason for this behaviour is the discontinuity at $t = 1$. For those timesteps that divide an interval of length 1 into an integer number of steps, the integration will be stopped and restarted *exactly at* the discontinuity in the derivative of the right-hand side at $t = 1$. Therefore, the error bound (Eq. (7)) holds, since the method does not step across the discontinuity. Stopping and restarting at discontinuities is precisely what Hairer et al. (1993, pp. 197–198) recommends in strategy III discussed in Section 3.1 above, and in this sense, isolated discontinuities in the derivatives are easy to handle, if it is known *a priori* where they are.

Inspired by this result, we have designed a special-purpose version of the 4th-order Runge-Kutta integrator, specifically for this problem with a discontinuity at $t = 1$. It is identical to the regular one in every way, except that if $t < 1 < t + h$, it divides that step into two steps, of length $1 - t$ and $h - (1 - t)$, such that the integration is always stopped and restarted at $t = 1$.

The global error as a function of timestep for this special-purpose integrator is also shown in Fig. 2 (dashed line), and we observe that it follows very closely the expected $h^4$ scaling, until the point where roundoff error starts to dominate. The additional computational expense of the special-purpose integrator is completely negligible in this case, as it takes at most one additional step compared to the regular 4th-order Runge-Kutta method, but as we see, it can increase the accuracy by several orders of magnitude. In the next section, we apply this idea to variable-timestep integrators for trajectory calculation in interpolated vector fields.

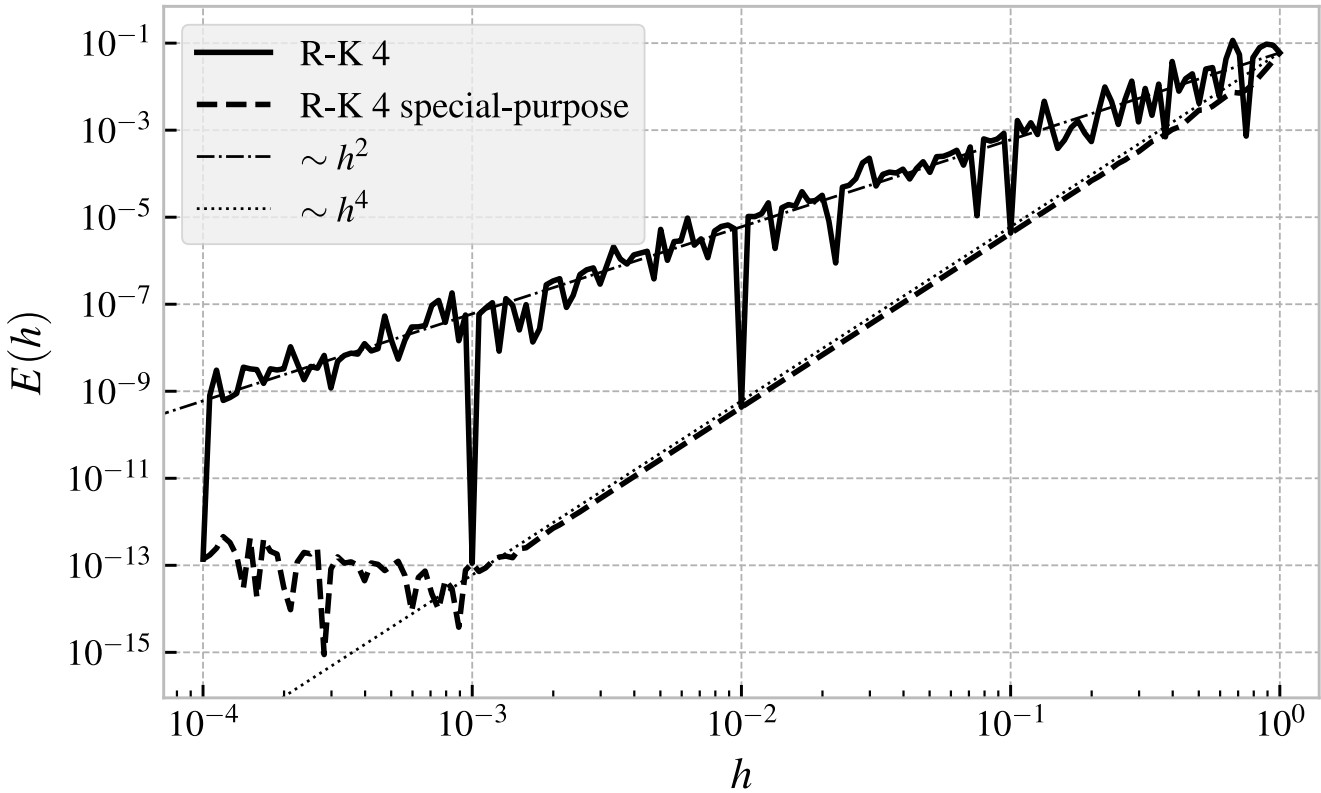

**Figure 2.** Global error in the numerical solution of the initial value problem given by Eq. (16), at $t_N = 2$. The solutions have been calculated with 161 different timesteps, $h$, logarithmically spaced from $10^{-4}$ to 1, using the 4th-order Runge-Kutta integrator, and a special-purpose modification of the same, that stops and restarts the integration exactly at the discontinuity at $t = 1$. The two thin lines are included to indicate the order of convergence, and are proportional to $h^2$ (dash-dotted line) and $h^4$ (dotted line).

### 3.3 Special-purpose integrators for interpolated velocity fields

In terms of the three strategies for dealing with discontinuities (see Section. 3.1) we will investigate a hybrid approach in
275 this paper. We will use information about the location of the discontinuities in the time dimension (strategy III), and leave
the error control routine to deal with the problem in the spatial dimensions (strategy I). The reason for this choice is mainly
pragmatic: For a particle trajectory, $\mathbf{x}(t)$, time is the independent variable, and it is very easy to stop and restart integration
at "cell boundaries" in the time direction. Doing the same in the spatial dimensions requires detection of boundary crossings,
dense output from the integrator, and a bisection-scheme to identify the time at which the boundary is crossed (Hairer et al.,
280 1993, pp. 188–196).

We will take as our starting point variable-timestep Runge-Kutta methods, as these are commonly used and generally quite
efficient, and the timestep adjustment routine outlined in Section 2.3. We then modify the timestep adjustment routine to make
sure the integration is always stopped and restarted at a cell boundary in time. We assume that the input data is given as





snapshots of a vector field at a list of known times, $T_i$. Depending on the degree of interpolation, the (higher) partial derivatives
of the interpolated vector field along the time dimension will thus have discontinuities at times $T_i$.

The variable timestep integrator calculating the trajectory will make steps, from position $\mathbf{x}_n$, at time $t_n$, to position $\mathbf{x}_{n+1}$, at
time $t_{n+1} = t_n + h_n$. Then, if we have $t_n < T_i < t_n + h_n$, for any $T_i$, i.e., if the integration is about to step across a discontinuity
in time, the timestep, $h_n$, is adjusted such that

$$h_n = T_i - t_n. \tag{18}$$

After that, integration and error control proceeds as normal. If $h_n$ is set to $T_i - t_n$, then a step to that time is calculated. The
error is then checked, as described in Section 2.3. If the error is found to be too large according to the selected tolerance, the
step is rejected, $h_n$ is further reduced, and the step is attempted again. In the opposite case, the step is accepted, and time and
position is updated to $t_{n+1}$ and $\mathbf{x}_{n+1}$. At this point, the timestep is reset to the original value of $h_n$, to avoid the integration
proceeding with an unnecessarily short timestep after the discontinuity has been crossed. For any step that does not cross a
discontinuity in time, the integrator behaves exactly like the regular version.

## 4 Numerical Experiments

The aim of the numerical experiments is to investigate the practical implication of different combinations of interpolation and
integration schemes, and to compare the special-purpose integrators described in Section 3.3 with their standard counterparts,
as well as with fixed-step Runge-Kutta methods. In the following subsections, we describe the input data, and the setup used
to carry out the numerical experiments.

In order to allow the interested reader to reproduce our results, we provide the Fortran code used to run the simulations, the
ocean current data used, and the jupyter notebooks that were used to analyse the data. These can all be found on github[1], under
an open-source license. In order to reduce the file size of the current data, the extents of the original datasets were reduced,
and unused variables were deleted from the files. The domains of the reduced datasets are shown in Fig. 3. All the datasets
were originally downloaded on the netCDF format from the ocean and ice section of the THREDDS server of the Norwegian
Meteorological Institute[2].

### 4.1 Ocean Currents

The datasets used were obtained from the Norwegian Meteorological Institute, and were taken from the following model
setups:

- Arctic20km (20 km horizontal resolution, 1 h timestep),

- Nordic4km (4 km horizontal resolution, 1 h timestep),

- NorKyst800m (800 m horizontal resolution, 1 h timestep).

---

[1]github.com/nordam/ODE-integrators-for-Lagrangian-particles
[2]thredds.met.no/thredds/fou-hi/fou-hi.html





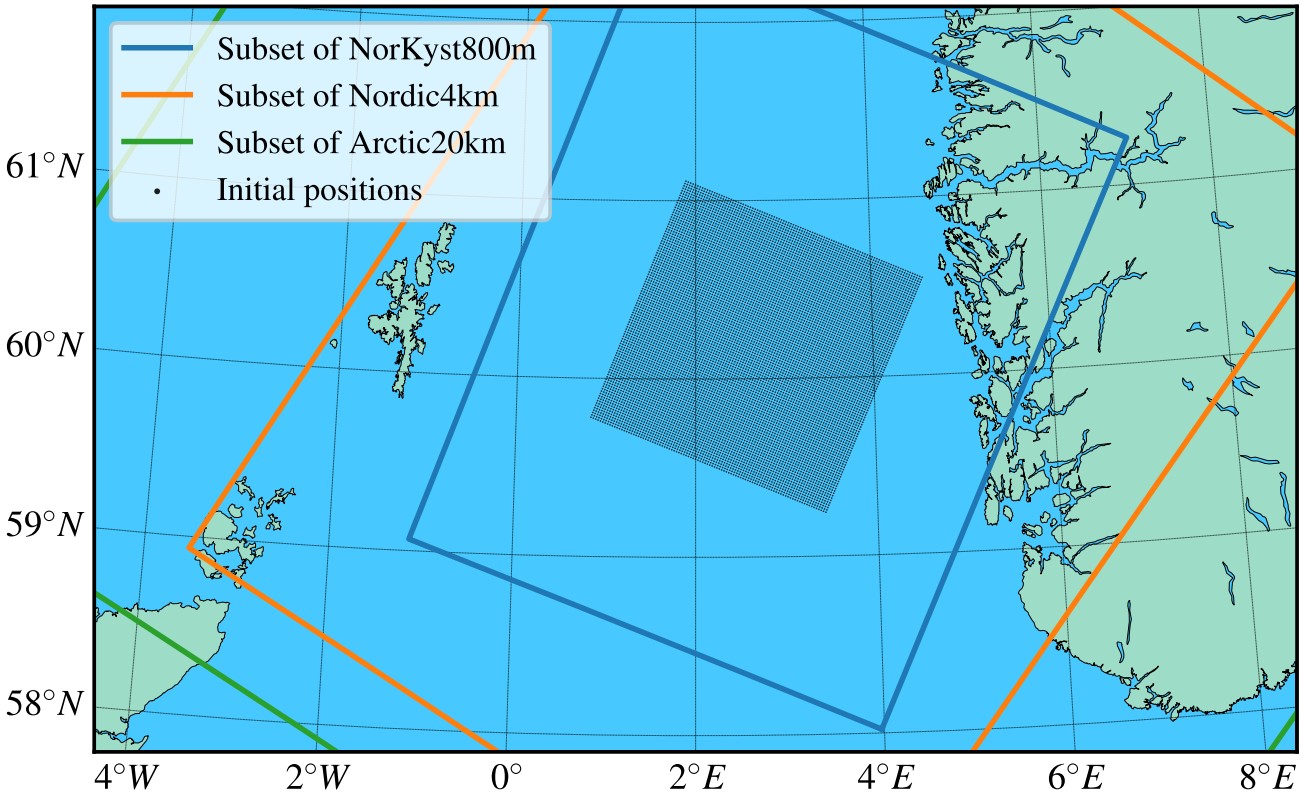

**Figure 3.** Map showing the outline of the three datasets considered, as well as the initial positions of the tracers used in the numerical experiments.

The dimensions of the datasets are $x$, $y$, $z$ and $t$, with the $xy$ plane defined in a polar stereographic projection, giving a regular (constant spacing) quadratic grid in the horizontal plane. The current velocity field is provided as vector components

on the $xy$ basis (as opposed to, e.g., an East-North basis). In our simulations, we track particle positions in meters, using the $xy$ coordinate system of the polar stereographic projection of the datasets. This allowed us to use the vector components directly from the datasets, with no rotation or other conversion. All error measurements are calculated from Euclidean distances in the $xy$ plane.

## 4.2   Initial conditions

The initial conditions for the trajectory calculations were chosen to be $100 \times 100$ points off the coast of Norway, placed on a regular quadratic grid with grid spacing of $1600\,\mathrm{m}$, as shown in Fig. 3. The same initial conditions were used for all three datasets. Roughly the easternmost half of the initial positions are within the Norwegian coastal current (see, e.g., Sætre (2005)), and are predominantly transported northward along the coast. The trajectories were started at midnight on February 8, 2017, and





integrated for 72 hours. All the particles remain inside the smallest domain (the $800\,\mathrm{m}$ resolution setup, see Fig. 3) throughout
this period.

### 4.3  Reference solutions

As we wish to estimate the global error of our numerical solutions, when the true solutions are unknown, we need to establish highly accurate numerical solutions for all $10\,000$ initial conditions, to use as a reference. Reference solutions must be established for each dataset, as they will in general give different transport. Additionally, reference solutions must also be
established for each interpolation scheme, as they will also in general give different trajectories. Hence, for three datasets and three interpolation schemes, we need nine different sets of reference solutions.

We point out that we here talk about reference solutions in a purely numerical sense, as the most mathematically accurate solution of the initial-value problem given by an initial position and a discrete velocity field with a specified interpolation scheme. Which of the datasets and interpolation schemes that most accurately reproduce the trajectories of true Lagrangian
drifters in the ocean is a different question, outside the scope of this investigation.

For numerically obtained reference solutions to be useable in calculating error estimates, they need to be significantly more accurate than any of the numerical solutions that are to be evaluated. As an example, consider a fixed-step integrator, and let the numerical solution at time $t_N$, calculated with a timestep $h$, be $\mathbf{x}_N(h)$, and let the true (but usually unknown) solution at time $t_N$ be $\mathbf{x}(t_N)$. Furthermore, assume that a reference solution $\mathbf{x}_N(h_{\mathrm{ref}})$ has been calculated with a very short timestep $h_{\mathrm{ref}}$.
Then, the error in the reference solution, relative to the true (but unknown) solution, is given by

$$E_{\mathrm{ref}} = \mathbf{x}_N(h_{\mathrm{ref}}) - \mathbf{x}(t_N). \tag{19}$$

Similarly, the error in a solution calculated with a longer timestep, $h$, is

$$E(h) = \mathbf{x}_N(h) - \mathbf{x}(t_N). \tag{20}$$

When we estimate the error by purely numerical means, we do not know the true solution, $\mathbf{x}(t_N)$. Instead, we use the reference
solution in place of the true solution, and calculate an estimate of the error, given by

$$\begin{aligned}\bar{E}(\Delta t) &= \mathbf{x}_N(h) - \mathbf{x}_N(h_{\mathrm{ref}}), \\ &= E(h) - E_{\mathrm{ref}}.\end{aligned} \tag{21}$$

Hence, we see that the numerical estimate, $\bar{E}(h)$, of the global error, is only a good estimate if $E_{\mathrm{ref}} \ll E(h)$.

To verify that the errors in the reference solutions are indeed much smaller than any of the other errors we wish to estimate, we consider the convergence of the numerically estimated error. The details of the analysis to identify reference solutions are
shown in Appendix A. We found that the most accurate solutions were obtained with the 4th-order Runge-Kutta integrator, using a fixed, short timestep. The timestep that yielded the most accurate solutions varied, depending on the dataset and the order of interpolation. The results are given in Table A2.



## 4.4 Implementation

To allow easy testing of different combinations of datasets, interpolators and integrators, in a setting relevant for marine
transport applications, a simple Lagrangian particle transport code was written in Fortran. All the integrators were implemented
as described in Sections 2.2, 2.3, and 3.3, and in the references given. The netCDF library for Fortran[3] was used to read
ocean current data, and interpolation was done using the library bspline-fortran[4] (Williams, 2018). Our implementations of the
different integrators, and all the code used to run the simulations, are freely available on github[5].

At the start of the simulations, subsets of the ocean current datasets were loaded from file. The horizontal extent of the
subsets are shown in Fig. 3, along with the initial positions of the particles. We used only the surface layer of the datasets, and
data spanning 5 days. The subsets were selected to cover the entire simulation period in time, and the entire horizontal extent
of the particle trajectories, with some padding on all sides to avoid edge-effects in the spline interpolation. Data points that
were on land were set to 0 current velocity. No special steps were taken to handle the coastline, although the initial conditions
were chosen to avoid particles getting stuck in land cells.

The data were passed to the `initialize` method of the derived type `bspline_3d` from the bspline-fortran library, along
with the parameter to select the order of interpolation (note that the order of a spline is 1 plus the polynomial degree, meaning
the order is 2 for linear interpolation, 4 for cubic splines, and 6 for quintic splines). The $x$ and $y$ components of the current
velocity vectors were interpolated separately, and the order of interpolation was always the same along all three dimensions
$(x, y, t)$.

We note that this approach constructs a single, global interpolation object, that is used throughout the simulation. It is also
possible to construct local spline interpolations using only the smallest required number of points, surrounding the location
where the function is to be evaluated ($2 \times 2 \times 2$ points for linear interpolation, $4 \times 4 \times 4$ for cubic splines, and $6 \times 6 \times 6$ for
quintic). However, this creates additional discontinuities in the derivatives of the right-hand side when switching from one local
interpolator to the next, as discussed by, e.g., Lekien and Marsden (2005).

During the simulations, the trajectory of each particle was calculated independently of all others. For the variable-step
integrators, this means that each particle had its own timestep. It is also possible to apply the variable-step integrators to all
particles simultaneously, with the same timestep. However, due the local variability of the ocean currents, it seemed more
reasonable to treat the particles individually, allowing the variable-step integrators to adapt to local conditions for each particle.

## 5 Results and Discussion

The main results are presented as a work-precision diagram, in Fig. 4. The figure shows the median relative global error
over all 10 000 particles, as a function of number of evaluations of the right-hand side of the ODE (including rejected steps).
The relative global error is calculated as the normalised distance between the endpoint of each trajectory, and that of the

---

[3] www.unidata.ucar.edu/software/netcdf/docs-fortran/

[4] github.com/jacobwilliams/bspline-fortran

[5] github.com/nordam/ODE-integrators-for-Lagrangian-particles



corresponding reference solution (see Section. 4.3 and Appendix A). Number of evaluations of the right-hand side was chosen as a measure of work, as it is more objective than the runtime of the simulation, which would depend on the particular machine

used to run the simulations, and also be more susceptible to somewhat random variations.

We note that higher-order interpolation is more computationally costly than lower order interpolation. For the simulations done with the fixed-step 4th-order Runge-Kutta integrator, the simulations with cubic spline interpolation took on average four to five times longer than those with linear interpolation, and the simulations with quintic spline interpolation took on average three to four times longer than those with cubic spline interpolation. As an example, calculating the trajectories of

10000 particles, for 72 hours, with a 10 minute timestep with the 4th-order Runge-Kutta integrator, took 11 seconds with linear interpolation, 51 seconds with cubic interpolation, and 177 seconds with quintic interpolation. The numbers were essentially the same for all three datasets ($800\,\mathrm{m}$, $4\,\mathrm{km}$ and $20\,\mathrm{km}$). These times cover only the trajectory calculation itself, not file I/O or the construction of the global interpolator object.

The fixed-step integrators were run with the range of timesteps shown in Table 1. Note that all of these steps evenly divide

the $3600\,\mathrm{s}$ interval of the data, making sure that the integration is always stopped and restarted at a cell boundary in the time dimension (see discussion in Section 3).

**Table 1.** Timesteps and tolerances used in the numerical experiments.

| | |
|---|---|
| Timesteps [s] | 120, 180, 300, 450, 600, 900, 1200, 1800, 3600. |
| Tolerances | $10^{-4}$, $10^{-5}$, $10^{-6}$, $10^{-7}$, $10^{-8}$, $10^{-9}$, $10^{-10}$, $10^{-11}$, $10^{-12}$, $10^{-13}$. |

The tolerances used with the variable-step integrators are also shown in Table 1, with $T_A = T_R$ (see Section 2.3). Note that in the coordinate system used, the particle positions are all of the order $10^6\,\mathrm{m}$, meaning that the relative tolerance dominates in practice (see Eq. (10)). Both the regular and the special-purpose variable-step integrators were used with the same tolerances,

but we note that the special-purpose integrators are by design unable to take steps longer than the interval on which the data is given. Hence, for the higher tolerances (allowing larger errors), the special-purpose integrators would default to fixed-step integration with a timestep of $3600\,\mathrm{s}$ (for the datasets used here).

We observe from Fig. 4 that the most efficient choice of integrator, in the sense of fewest evaluations of the right-hand side for a given accuracy, depends on the desired accuracy, the order of the interpolation, and the spatial resolution of the dataset.

We will discuss these points in turn.

### 5.1 Fixed step integrators

Variable-step integrators are normally the most efficient choice for general ODE problems. However, we see that for finding tracer trajectories from interpolated velocity fields, fixed-step integrators are in some cases a better choice than regular variable-step methods. Considering for example cubic spline interpolation (Fig. 4, middle row), we see that 4th-order Runge-Kutta



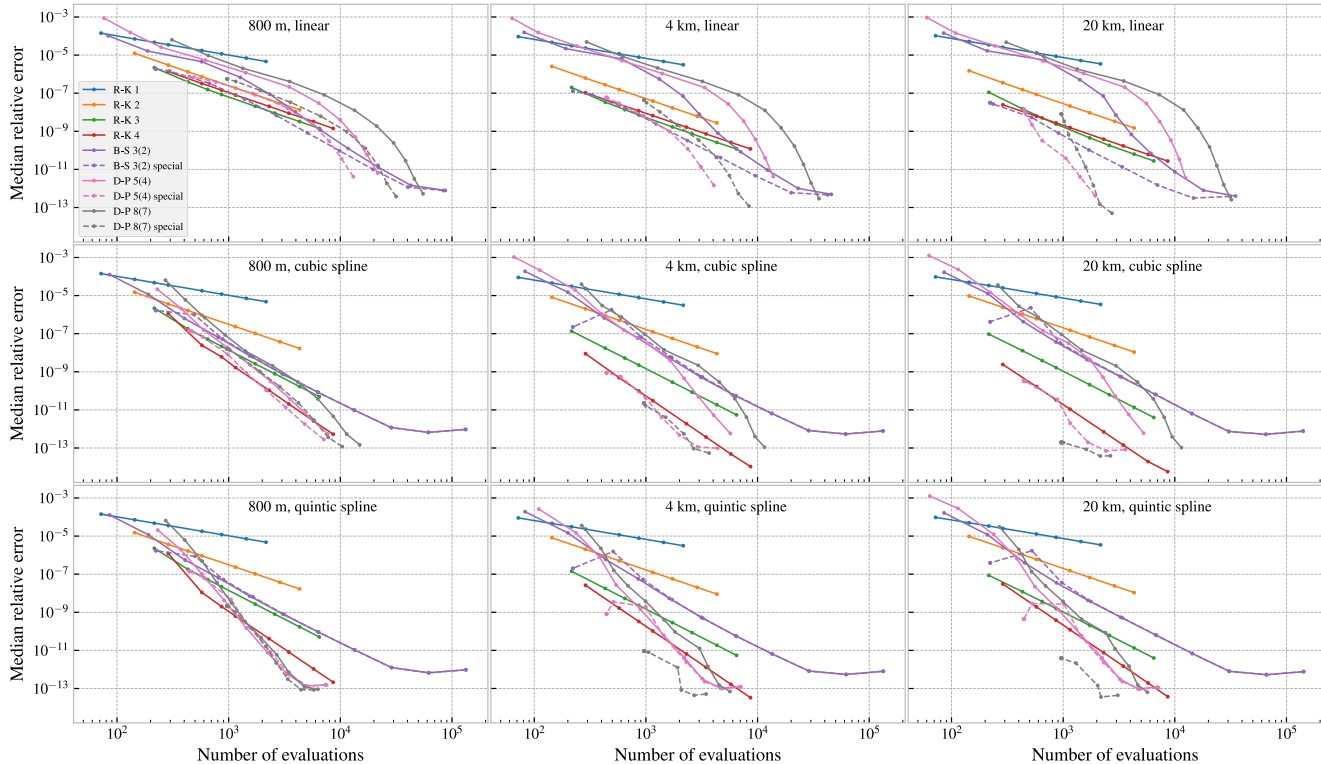

**Figure 4.** Relative global error (relative to the reference solution) as a function of number of evaluations of the right-hand side. Note that the special-purpose integrators are (by design) unable to make longer steps than the interval on which the data are provided. This means some of the simulations with higher tolerance (allowing larger errors) have in practice defaulted to fixed-step simulation with a timestep of 3600 s, making several of the data points identical. This is most readily observed for the special-purpose Dormand-Prince 8(7) integrator, in the lower right panel.

almost always gives better accuracy for the same amount of work, relative to all three regular variable-step integrators. The only exception is for very small errors, for the 800 m dataset, where Dormand-Prince 5(4) has a small advantage. Similarly for linear interpolation (Fig. 4, top row), the 3rd- and 4th-order fixed-step methods outperform the regular variable-step methods, except if very small errors are required.

The special-purpose variants of the variable-step integrators, particularly Dormand-Prince 5(4) and 8(7), perform better than the fixed-step methods in most cases, though not always by a large margin. The reason for the relatively strong performance of the fixed-step integrators is that the chosen timesteps evenly divide the 3600 s intervals of the datasets. Hence, the fixed-step integrators will stop and restart integration at the discontinuities in time, just like the special-purpose integrators (see Section 3.3). For an illustration of the effect of choosing timesteps that do *not* evenly divide the temporal grid spacing of the dataset, see Nordam et al. (2017, Fig. 18).





We also note that for the case of linear interpolation, the 3rd-order Runge-Kutta integrator actually performs slightly better than the 4th-order, particularly for the smaller errors. The reason for this is that the lack of continuous derivatives means the 4th-order method does not achieve 4th-order convergence. As the 3rd-order method uses one fewer evaluation of the right-hand side per step, it therefore has an advantage in terms of computational effort. It is also worth pointing out that the 2nd-order Runge-Kutta method considered here, known as the explicit trapezoid method, has the advantage that it uses no intermediate

points in time. Since it only evaluates the right-hand side at times $t_n$ and $t_{n+1}$, it is possible to dispense with interpolation in time entirely if one selects the integration timestep, $h$, to be equal to the temporal grid spacing of the data. Note that this requires reasonably high temporal resolution of the dataset, which may not always be practical.

### 5.2    Variable-step integrators

As a background for discussing the effect of horizontal resolution on our results, we recall that all the three datasets used

have a temporal resolution of 1 hour. This means that the particle trajectories will cross a cell boundary in the time-dimension (and thus a discontinuity in the (higher) derivatives of the right-hand side) every hour. The average current speed for the time and area studied is approximately $0.2 \, \mathrm{m/s}$ in all three datasets. Hence, we find that a particle that moves in the velocity field defined by the dataset at $800 \, \mathrm{m}$ spatial resolution will cross a spatial cell boundary approximately once every hour on average. For the dataset with $4 \, \mathrm{km}$ resolution, this will only happen 1/5th as often, and for the $20 \, \mathrm{km}$ dataset, only 1/25th as often.

These are only crude estimates, but we can nevertheless conclude that for the low-resolution datasets, the errors picked up at the discontinuities in time will be more important than those in space, while for the high-resolution ($800 \, \mathrm{m}$) dataset, the two will be of similar importance.

     Looking at the results presented in Fig. 4, we find that they support these observations. Considering first the case of linear interpolation, we see that for the $20 \, \mathrm{km}$ dataset (Fig. 4, upper right panel), there is a considerable (several orders of magnitude)

reduction in error in the special-purpose integrators, compared to the regular variable-step integrators for a given number of evaluations. Recall that the only difference between these is that the special-purpose integrators stop and restart the integration at every cell boundary along the time dimension (see Section 3.3). For the $800 \, \mathrm{m}$ dataset (Fig. 4, upper left panel) on the other hand, there is less (up to about an order of magnitude) difference between the regular and special variable-step integrators. This is presumably because the discontinuities in time do not dominate the error as much in this case.

Looking next at the results for cubic spline interpolation (Fig. 4, middle row), we notice that the results for the regular and special-purpose versions of the Bogacki-Shampine 3(2) integrator are now practically identical. For the Dormand-Prince 5(4) and 8(7) integrators, the special-purpose variants are far more accurate than the standard counterparts. This is particularly true for the $4 \, \mathrm{km}$ and $20 \, \mathrm{km}$ datasets, where the difference is several orders of magnitude.

     Presumably, the reason why the standard and special-purpose variants of the Bogacki-Shampine 3(2) integrator give more

or less identical results for cubic interpolation is the smoothness of the velocity field. It seems the interpolated field is now sufficiently smooth that the method is now third-order consistent. Strictly speaking, this is unexpected. A cubic spline interpolation will have continuous second derivatives, and discontinuous third derivatives. This means that the Bogacki-Shampine





3(2) integrator can indeed be expected to be second-order consistent, but the conditions for the third-order consistency are not satisfied.

Using quintic spline interpolation (Fig. 4, bottom row), the special-purpose variant of the Dormand-Prince 8(7) integrator performs better than all the other methods by at least an order of magnitude. We also find that the results for the regular and special-purpose versions of the Dormand-Prince 5(4) integrators are more or less identical. As above, this was not entirely expected, since a quintic spline has only four continuous derivatives, not the five that are theoretically required for the local error of a 5th-order method to be bounded by Eq. (7).

To understand the large differences in number of function evaluations between the standard and the special-purpose integrators, we look at the fraction of rejected steps. For the different integrators and interpolators, and a fixed tolerance of $T_A = T_R = 10^{-10}$, these fractions are given in Table 2. Rejected steps represent wasted computational effort, since a rejected step requires as many evaluations of the right-hand side of the ODE as an accepted step, without advancing the integration.

The results shown in Table 2 further support the conclusions we drew from Fig. 4 above. For those cases where the order

of interpolation is less than the theoretical requirements of the integrator, the special-purpose integrators significantly reduce the fraction of rejected steps. The difference is also largest for the $20\,\mathrm{km}$ dataset, as discussed previously. This can be seen particularly for the Dormand-Prince 8(7) integrator with cubic and quintic interpolation, where the rejected fraction falls to almost nothing for the special-purpose variant. The same, but to a lesser degree, is seen for the Dormand-Prince 5(4) integrator, with linear and cubic interpolation. On the other hand, for the Bogacki-Shampine 3(2) integrator, with cubic and quintic

interpolation, we see that there is essentially no difference between the regular and special variants, as the velocity field is sufficiently smooth for the error control routine not to detect any increased local error at the boundary crossings.

The largest improvement in accuracy for the special-purpose integrators is thus seen with linear interpolation, but they can also be advantageous with cubic interpolation. With quintic interpolation, only the special-purpose (8)7 integrator has an advantage over its regular counterpart. However, the relative error of the special (8)7 method with quintic interpolation

is comparable to the (5)4 method with cubic interpolation. While the solutions will be different with different interpolation schemes, it is possible that overshooting due to a high order interpolation method without any additional accuracy implies that the (8)7 method is not a good choice for Lagrangian oceanography. Note also that the quintic interpolation scheme is 3-4 times as computationally expensive as the cubic, for each evaluation of the right-hand side.

### 5.3   Summary

We have seen that the special-purpose integrators are more efficient than their regular counterparts in almost all cases, and sometimes they deliver several orders of magnitude improvement in accuracy at the same computational cost. There are two different effects that give the special-purpose integrators their advantage in accuracy and efficiency. The first is that they stop and restart integration exactly at the discontinuities in time, which avoids picking up local errors unbounded by Eq. (7) at those points. The second effect is that they avoid many rejected steps by stopping at the discontinuity, instead of trying to step across.

The regular variable-step integrators will frequently try to step across a discontinuity, only to find that the estimated local error is too large, such that the step must be rejected and retried with a shorter timestep. This process will continue until a

**Table 2.** Fraction of steps rejected, averaged over all 10000 trajectories, with a duration of 72 hours, for each combination of interpolation scheme and variable-stepsize integrator, for all three datasets, and a fixed tolerance of $T_A = T_R = 10^{-10}$ (see Section 2.3).

| Resolution | Integrator | Linear | Cubic | Quintic |
|---|---|---|---|---|
| 20 km | B-S 3(2) | 0.334 | 0.017 | 0.018 |
| 20 km | B-S 3(2) special | 0.067 | 0.016 | 0.018 |
| 20 km | D-P 5(4) | 0.588 | 0.486 | 0.251 |
| 20 km | D-P 5(4) special | 0.084 | 0.113 | 0.156 |
| 20 km | D-P 8(7) | 0.608 | 0.558 | 0.482 |
| 20 km | D-P 8(7) special | 0.152 | 0.000 | 0.000 |
| 4 km | B-S 3(2) | 0.309 | 0.023 | 0.024 |
| 4 km | B-S 3(2) special | 0.095 | 0.022 | 0.024 |
| 4 km | D-P 5(4) | 0.587 | 0.436 | 0.247 |
| 4 km | D-P 5(4) special | 0.289 | 0.115 | 0.158 |
| 4 km | D-P 8(7) | 0.609 | 0.554 | 0.394 |
| 4 km | D-P 8(7) special | 0.379 | 0.012 | 0.019 |
| 800 m | B-S 3(2) | 0.266 | 0.016 | 0.016 |
| 800 m | B-S 3(2) special | 0.161 | 0.016 | 0.016 |
| 800 m | D-P 5(4) | 0.580 | 0.294 | 0.217 |
| 800 m | D-P 5(4) special | 0.490 | 0.159 | 0.152 |
| 800 m | D-P 8(7) | 0.615 | 0.468 | 0.237 |
| 800 m | D-P 8(7) special | 0.545 | 0.269 | 0.124 |

timestep is found that is short enough to allow the discontinuity to be crossed with an error that stays within the tolerance. As we see from the results in Table 2, this can lead to a large fraction of rejected steps. Also, recall that the regular variable-step integrators have no information about the location of the discontinuities in time, which means that the probability of stopping

and restarting the integration exactly at a discontinuity is essentially zero. For further details, see the discussion in Section 3, as well as Hairer et al. (1987, p. 181) and Hairer et al. (1993, pp. 197–198).

## 6 Conclusions

The most striking conclusion from the results presented above, is that the special-purpose integrators in many cases deliver several orders of magnitude more accurate results, at no additional cost. Alternatively, they can deliver the same accuracy as

standard methods, with highly reduced computational effort. This is particularly true for linear and cubic interpolation, and the $4\,\mathrm{km}$ and $20\,\mathrm{km}$ datasets. The increased efficiency of these integrators should be particularly relevant for long-term simulations, such as studies of global transport of plastics or global climate simulations.





Going more into details, we find that the most efficient choice of integrator depends on the resolution of the dataset, the degree of interpolation, and the desired accuracy. Looking at cubic interpolation (Fig. 4, middle row), we find that the fixed-

step 4th-order Runge-Kutta method is in most cases a more efficient choice than a standard variable-step integrator (provided the timestep is selected to evenly divide the interval of the dataset). The difference varies with the resolution of the dataset and the required accuracy, but in some cases the error is two orders of magnitude smaller for the 4th-order Runge-Kutta than the regular Dormand-Prince 5(4) method. This is an interesting result, given that the combination of cubic interpolation and a variable-step integrator such as Dormand-Prince 5(4) or Runge-Kutta-Fehlberg (Hairer et al., 1993, p. 177) appears to be a

popular choice. In the case of the $20\,\mathrm{km}$ dataset, and to a lesser extent for the $4\,\mathrm{km}$ dataset, additional accuracy can be gained by switching to a special-purpose variant of the Dormand-Prince integrators.

For linear interpolation, we find that if very small errors are required, the regular variable-step integrators perform better than the fixed-step methods, and in particular the Bogacki-Shampine 3(2) integrator. The specal-purpose variable-step methods achieve notable improvements, often being several orders of magnitude more precise. For less strict requirements, the 3rd-order

Runge-Kutta method appears to be the best choice. However in all cases, there is a considerable improvement in accuracy with the special-purpose integrators relative to the regular variable-step methods.

For quintic spline interpolation, the optimal choice of interpolator again depends on the application. If very small errors are required, the Dormand-Prince 5(4) method appears to be the best performer, or alternatively the special-purpose variant of Dormand-Prince 8(7). If larger errors are acceptable, the 4th-order Runge-Kutta method seems to be the better choice.

It is interesting that if an appropriate fixed step is chosen (i.e., a step that divides the interval between discontinuities in time), the 4th-order Runge-Kutta method is more efficient than the regular Dormand-Prince (5)4 method for all ocean model resolutions. This is true for any interpolation scheme and accuracies, except linear and quintic interpolations when very small errors are desired. The 4th-order method with a good choice of time step also performs well relative to the special-purpose 5(4) method although the latter may significantly outperform the former with linear and cubic interpolations. The strong

performance of the 4th-order Runge-Kutta with all resolutions and interpolation schemes makes it a good practical choice.

To conclude, we have investigated the accuracy of trajectory calculation with 10 different ODE integrators, for 9 different combinations of current data resolution and order of interpolation. We find that the optimal choice of integrator depends on the interpolation, the resolution, and the required accuracy. In some cases, the most efficient integrator is not the most popular choice in the literature.

We have designed and investigated special-purpose variants of the regular variable-step integrators. Only minimal changes to the code is required, to ensure that integration is always stopped and restarted at discontinuities in time. With this change, these special-purpose integrators can in some cases increase the accuracy by many orders of magnitude, for the same amount of computational effort. For applications requiring large numbers of trajectories, such as LCS calculations, or for long-term transport calculations, the added accuracy of the special-purpose methods should allow significant reductions in computational

expense.





*Code and data availability.* The code used to run the simulations and analyse the results, as well as the three different ocean current datasets, can be found at github.com/nordam/ODE-integrators-for-Lagrangian-particles.

## Appendix A: Reference solutions

**Table A1.** Timesteps and tolerances used in establishing reference solutions.

| Timesteps [s] | 3600, 1800, 1200, 900, 450, 300, |
| --- | --- |
| | 180, 120, 60, 30, 10, 5, 2, 1. |
| Tolerances | $10^{-4}$, $10^{-5}$, $10^{-6}$, $10^{-7}$, $10^{-8}$, $10^{-9}$, $10^{-10}$, $10^{-11}$, |
| | $10^{-12}$, $10^{-13}$, $10^{-14}$, $10^{-15}$, $10^{-16}$, $10^{-17}$. |

In order to establish highly accurate reference solutions, which are needed to estimate the error when the true solutions are
unknown, an expanded set of timesteps and tolerances were investigated. These are given in Table A1. For each timestep in the expanded set, a solution was calculated with the 4th-order Runge-Kutta method, and for each tolerance in the expanded set, a solution was calculated with the Dormand-Prince 8(7) method, using both the regular and the special-purpose variant. This was done for each of the three datasets, and for each of the three orders of interpolation.

### A1   Roundoff error and truncation error

Every step with a numerical ODE integrator contains some error. The *truncation error* stems from approximations that are made in constructing the integrator, and decreases with timestep. The *roundoff error* comes from the finite-precision representation of numbers on a computer, and is independent of the timestep. Due to numerical roundoff error, one can not simply assume that the shortest timesteps or smallest tolerance will always give the most accurate answer. As the number of steps increase, the roundoff error will eventually become larger than the truncation error, at which point no accuracy is gained by reducing the
stepsize further.

Loosely speaking, a double precision floating point number can store approximately 16 significant digits, and any numerical operation should be thought of as introducing a roundoff error in the least significant digit (Press et al., 2007, p. 10). This means that any step with an ODE integrator unavoidably introduces a relative error of approximately $10^{-16}$. As the timestep is reduced, the numbers of steps increase, and eventually the net contribution of the added roundoff errors will dominate. An
example of this can be seen in Fig. 2, where the error of the special-purpose method decreases down to about $10^{-13}$, whereafter it begins to increase with further reduction of the timestep.

### A2   Finding the most accurate solutions

In order to establish the most accurate solutions, we compare the 4th-order Runge-Kutta solutions obtained with very short timesteps, and Dormand-Prince 8(7) solutions with very small tolerances. We let the 4th-order Runge-Kutta solutions obtained





with timestep $h$ be given by $\mathbf{x}_N(h)$, and the Dormand-Prince 8(7) solutions obtained with relative tolerance $T_R$ (see Section 2.3)
be given by $\mathbf{x}_N(T_R)$. We also let the (unknown) true solution be given by $\mathbf{x}(t_N)$. Then we consider the relative difference
between these numerical solutions, $\Delta(h, T_R)$, given by

$$\Delta(h, T_R) = \frac{|\mathbf{x}_N(h) - \mathbf{x}_N(T_R)|}{|\mathbf{x}_N(T_R)|} \tag{A1a}$$

$$= \frac{\left|\big(\mathbf{x}_N(h) - \mathbf{x}(t_N)\big) - \big(\mathbf{x}_N(T_R) - \mathbf{x}(t_n)\big)\right|}{|\mathbf{x}_N(T_R)|}. \tag{A1b}$$

In Eq. (A1b), we have added and subtracted the true (but typically unknown) solution, $\mathbf{x}(t_N)$, highlighting that $\Delta(h, T_R)$ is
also equivalent to the difference in the global error of the fixed-step and variable step solutions (see Eq. 6).

To evaluate the accuracy of the numerical solutions, we first keep the tolerance, $T_R$, fixed, and we plot the median rela-
tive difference as a function of timestep, $h$. The result is shown in Fig. A1. We observe that for longer timesteps, the relative
difference, $\Delta(h, T_R)$, goes down with the timestep, $h$. Starting from the bottom row of Fig. A1, we observe that for quintic
interpolation, $\Delta(h, T_R)$ scales as $h^4$ (dashed lines). This is as expected, since a quintic spline has continuous partial deriva-
tives up to order four, as required for the 4th-order Runge-Kutta method to be guaranteed to deliver 4th-order accuracy (see
discussion in Sections 2.1 and 2.4, as well as Hairer et al. (1993, p. 157)). We also observe the same trend for cubic interpola-
tion (Fig. A1, middle row), while for linear interpolation (Fig. A1, top row), we find that the estimated error only goes down
proportional to $h^2$, due to the lack of continuous derivatives.

For shorter timesteps, we observe that the relative difference, $\Delta(h, T_R)$, flattens out and becomes constant. The interpretation
of this, in light of Eq. (A1b), is that for the shorter timesteps, $\Delta(h, T_R)$ is dominated by the error in the variable-step reference
solution, thus appearing to be constant with the timestep $h$. Based on this reasoning, we conclude that the most accurate
variable-step solutions are obtained with the special-purpose integrator, with a tolerance of $10^{-13}$, $10^{-14}$, or $10^{-15}$, depending
on the dataset and the order of interpolation.

Next, we do the opposite comparison, i.e., we use the 4th-order Runge-Kutta solutions as reference, keep the timestep fixed
and look at the relative difference, $\Delta(h, T_R)$, as a function of tolerance. The results are shown in Fig. A2. Starting from the
high tolerances, we observe that the relative difference first goes down as the tolerance is reduced. Then, in all cases except the
linearly interpolated 800 m dataset, the smallest estimated differences thereafter go up as the tolerance is reduced further. The
reason is that the error in the variable-step solutions goes down until at some point the accumulated roundoff errors begin to
dominate, and the error increases as the reduced tolerance leads to an increasing number of steps.

From Figs. A1 and A2 together, we conclude that the 4th-order Runge-Kutta solutions for short timesteps are the most
accurate solutions. As we can see from Eq. (A1b), we are essentially considering the absolute value of the difference in the
error of the fixed-step solution, and the error in the variable-step solution. Since $\Delta(h, T_R)$ (Fig. A1) appears constant with
timestep (for the shortest timesteps), we conclude that $\Delta(h, T_R)$ is dominated by the (relatively) large, constant error in the
variable-step solution, obscuring the small changes with timestep in the error in the fixed-step solution.

In order to further investigate the relative accuracy of the 4th-order Runge-Kutta solutions, we consider the change in the
solution between two different values of the timestep. First, we list all the timesteps in Table A1, such that $h_0 = 1\,\mathrm{s}$, $h_1 = 2\,\mathrm{s}$,





$h_2 = 5\,\mathrm{s}$, $h_3 = 10\,\mathrm{s}$, etc. Then we consider the quantity

$$\Delta_{\mathrm{RK4}}(h_i, h_{i+1}) = \frac{|\mathbf{x}_N(h_{i+1}) - \mathbf{x}_N(h_i)|}{|\mathbf{x}_N(h_i)|} \tag{A2a}$$


$$= \frac{\left|\big(\mathbf{x}_N(h_{i+1}) - \mathbf{x}(t_N)\big) - \big(\mathbf{x}_N(h_i) - \mathbf{x}(t_N)\big)\right|}{|\mathbf{x}_N(h_i)|} \tag{A2b}$$

As $h_i$ and $h_{i+1}$ become smaller, we expect $\Delta_{\mathrm{RK4}}(h_i, h_{i+1})$ to become smaller as well. Since the global error of a 4th-order Runge-Kutta method (for sufficiently smooth right-hand sides) is $\mathcal{O}(h^4)$, we see from Eq. (A2b) that

$$\Delta_{\mathrm{RK4}}(h_i, h_{i+1}) \sim \big(\mathcal{O}(h_{i+1}^4) - \mathcal{O}(h_i^4)\big). \tag{A3}$$

In Fig. A3, we plot $\Delta_{\mathrm{RK4}}(h_i, h_{i+1})$, as a function of $h_i$. For the linearly interpolated datasets, we observe that $\Delta_{\mathrm{RK4}}(h_i, h_{i+1})$

decreases proportionally to $h^2$, since the linearly interpolated right-hand sides are not sufficiently smooth to yield 4th-order convergence, and does not flatten out for small timesteps. Hence, we conclude that the solutions obtained with timestep $h = 1\,\mathrm{s}$ are the most accurate in this case.

With cubic and quintic interpolation, we see that $\Delta_{\mathrm{RK4}}(h_i, h_{i+1})$ goes down approximately as $h^4$, and eventually flattens out and increases a little for the shortest timesteps. As discussed previously, we interpret this to mean that the accumulated

roundoff errors begin to dominate. We find that the smallest difference is obtained with different timesteps for the different datasets. For example, for the $800\,\mathrm{m}$ resolution dataset, a timestep $h = 5\,\mathrm{s}$ appears to be the most accurate, while for the $20\,\mathrm{km}$ dataset, a timestep of $30\,\mathrm{s}$ appear to give better accuracy.

Based on the analysis described above, we have decided to use the 4th-order Runge-Kutta method to obtain the reference solutions used for the analysis in Section 5. For each dataset and order of interpolation, the reference timestep is chosen based

on Fig. A3, and the results are shown in Table A2.

**Table A2.** Timestep used with the 4th-order Runge-Kutta method, to obtain the reference solutions used in Section 5, for each order of interpolation and each dataset.

|        | 800 m | 4 km | 20 km |
|--------|-------|------|-------|
| Linear | 1 s   | 1 s  | 1 s   |
| Cubic  | 5 s   | 30 s | 30 s  |
| Quintic| 5 s   | 30 s | 30 s  |

As a final remark, we mention that it may seem surprising that we are able to obtain higher accuracy with the 4th-order Runge-Kutta method than with the Dormand-Prince 8(7) method. Three things are worth pointing out in this context. First, the timesteps considered here (see Table A1) all evenly divide the 1 hour step of the data, which means that a fixed-step method will always stop and restart the integration at the discontinuities in the time-direction (see discussion in Section 3.1). Second,

for the Dormand-Prince 8(7) method to work optimally, the right-hand side of the ODE should strictly have continuous partial derivatives up to order 8, which would require spline interpolation of degree 9. Finally, variable-step methods are generally



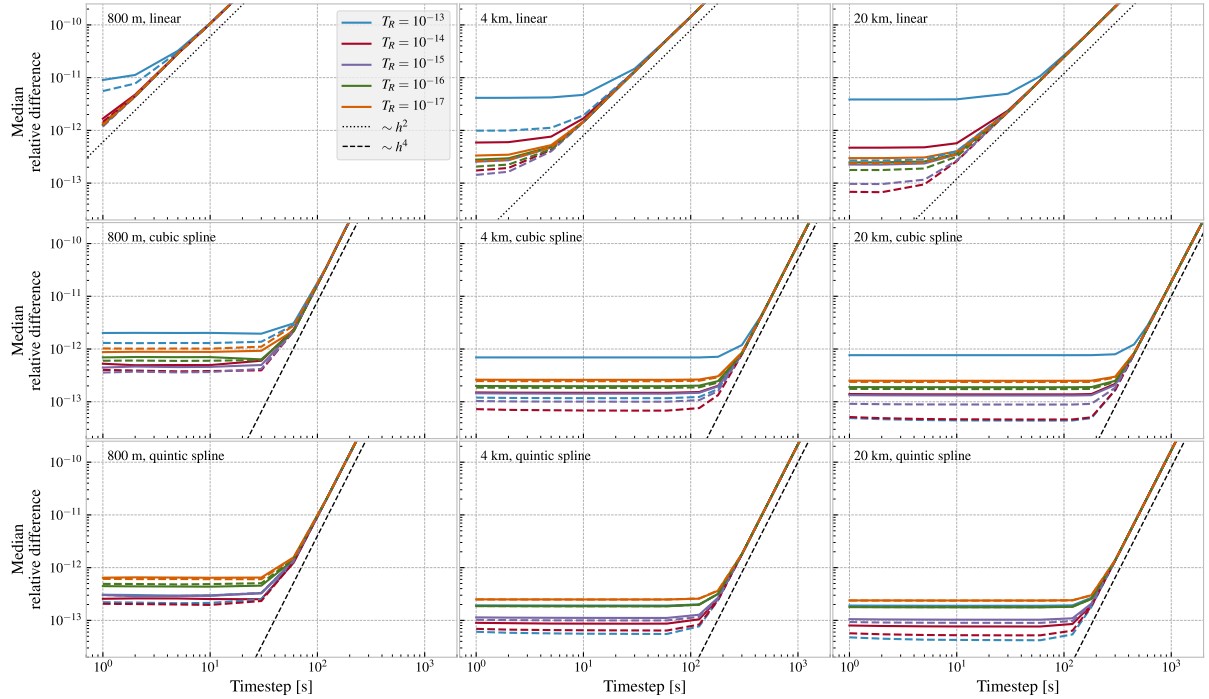

**Figure A1.** Median relative difference (Eq. (A1)) between the 4th-order Runge-Kutta solutions and the Dormand-Prince 8(7) solutions, as a function of the timestep for the Runge-Kutta method, and shown for different tolerances for the Dormand-Prince method. The regular Dormand-Prince 8(7) is shown as continuous lines, and the special-purpose variant as dashed lines.

preferred for their *efficiency*, not purely for their accuracy. As an example, consider the fifth-degree interpolated $800\,\mathrm{m}$ dataset. In this case, the presumed most accurate fixed-step solution, with $h = 5\,\mathrm{s}$ used $207\,360$ evaluations of the right-hand side, while the most accurate Dormand-Prince 8(7) solution, with a tolerance of $10^{-14}$, used $5805$ evaluations (including 17% rejected steps).

*Author contributions.* TN wrote the simulation code, and the first draft of the manuscript. Both authors participated in development of ideas, analysis of results, and writing of the final manuscript.

*Competing interests.* The authors declare that they have no competing interests.

*Acknowledgements.* The work of TN was supported in part by the Norwegian Research Council, through the project INDORSE (267793). TN would also like to thank his colleagues for many a good discussion in the SINTEF CoffeeLab.



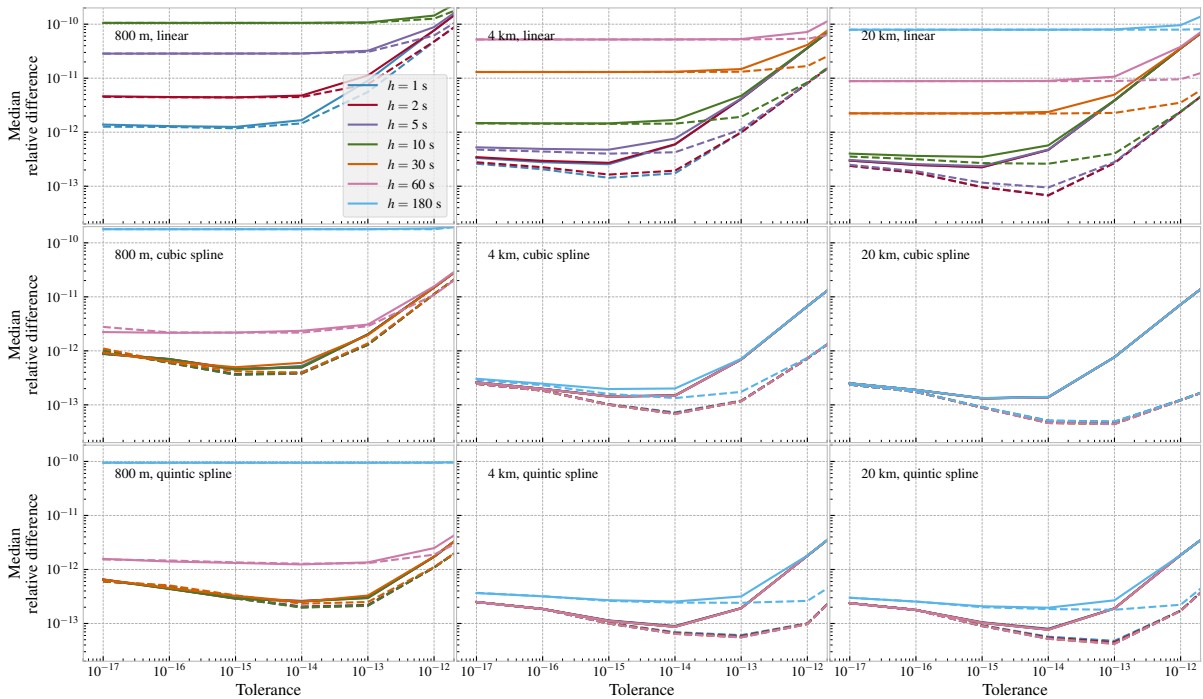

**Figure A2.** Median relative difference (Eq. (A1)) between the Dormand-Prince 8(7) solutions and the 4th-order Runge-Kutta solutions, as a function of the tolerance for the Dormand-Prince method, and shown for different timesteps for the Runge-Kutta method. The regular Dormand-Prince 8(7) is shown as continuous lines, and the special-purpose variant as dashed lines.

The work of RD was performed in support of the US Department of Energy's Fossil Energy, Oil and Natural Gas Research Program. It was executed by NETL's Research and Innovation Center, including work performed by Leidos Research Support Team staff under the RSS contract 89243318CFE000003. This work was funded by the Department of Energy, National Energy Technology Laboratory, an agency of the United States Government, through a support contract with Leidos Research Support Team (LRST). Neither the United States Government nor any agency thereof, nor any of their employees, nor LRST, nor any of their employees, makes any warranty, expressed or implied, or assumes any legal liability or responsibility for the accuracy, completeness, or usefulness of any information, apparatus, product, or process disclosed, or represents that its use would not infringe privately owned rights. Reference herein to any specific commercial product, process, or service by trade name, trademark, manufacturer, or otherwise, does not necessarily constitute or imply its endorsement, recommendation, or favoring by the United States Government or any agency thereof. The views and opinions of authors expressed herein do not necessarily state or reflect those of the United States Government or any agency thereof.

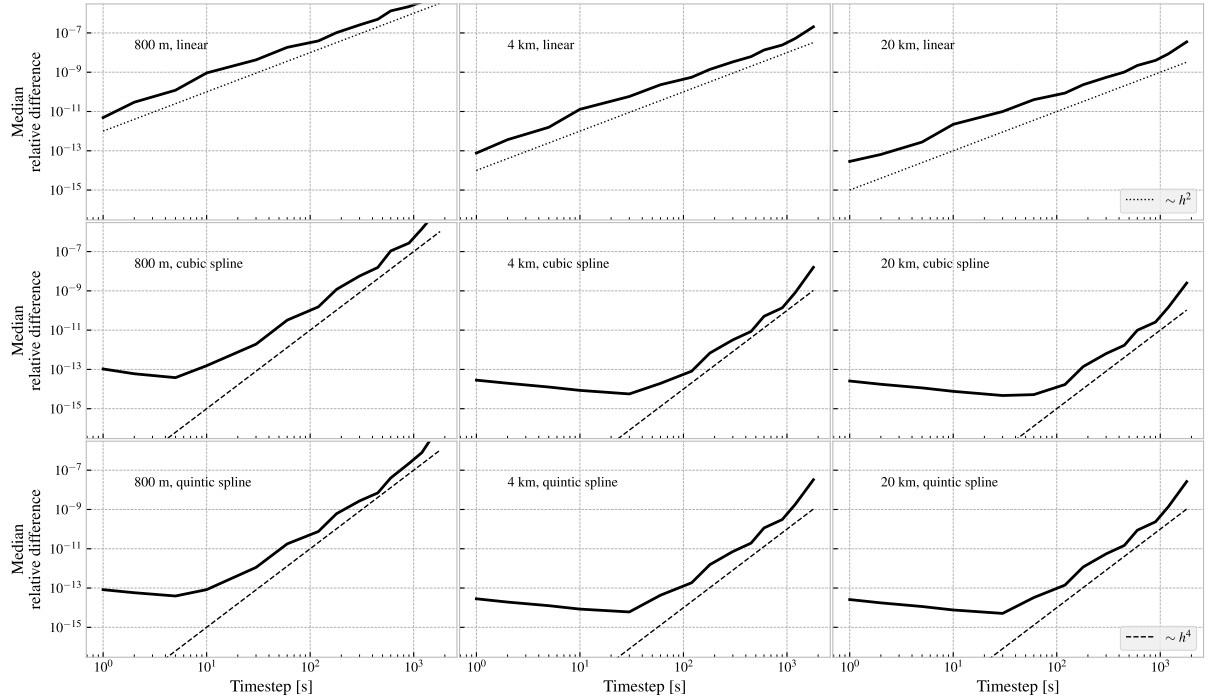

**Figure A3.** Median relative difference (Eq. (A2)) between two 4th-order Runge-Kutta solutions, obtained with different timesteps $h_i$ and $h_{i+1}$, using the list of timesteps in Table A1.

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
