# Peer review of "Numerical integrators for Lagrangian oceanography"

_Geoscientific Model Development, 2020_

## Referee Comment (RC1) · Anonymous Referee #1 · 8 Jul 2020

In this manuscript, the authors discuss the accuracy of different numerical integrators that are typically used in Lagrangian oceanography. They compare the efficiency and accuracy of a large set of these integrator algorithms in numerical flow fields with different spatial resolutions in the North Sea, off the coast of Norway.

The motivation of this manuscript is a really good one; many applications of Lagrangian oceanography choose an integrator without much idea about its efficiency and accuracy. This manuscript would therefore be very welcome to the field.

However, I feel that on a number of points, the manuscript could be further improved and clarified before I can recommend publication. In particular, I have the following major comments

1) The authors make the case for higher-order spatial interpolation, as if that is always better. However, there is no discussion at all about how the order of spatial interpolation is related to the order of the advection schemes within the OGCM from which the data is derived. Intuitively, I'd say that the most appropriate interpolation would be the scheme that most closely mimics the model advection scheme. I'd suggest the authors discuss this.

2) The authors also but then quickly step over the problem of interpolation near land. Higher-order interpolation would mean that the halo of land is further extended into the ocean (as very briefly mentioned in line 362). I feel that this should be given more discussion. How would this problem compare to the error that the 'consistency of order p of the numerical method is no longer satisfied when the derivatives are not continuous' (lines 222-223)

3) There is no discussion at all about the widely-used Analytical advection scheme of e.g. https://doi.org/10.5194/gmd-10-1733-2017. How does that scheme compare to the integrators discussed here?

4) The authors mention that they ignore diffusion (line 74). However, in most applications diffusion will be included in the computations. I wonder whether the errors caused by the finite-sized set in the Wiener process are not much larger than any errors in interpolation as discussed here. It would be good if the authors could comment on this.

5) I wonder why the authors don't test their method on (complex) flows where an exact solution is known. Quite a few of these flows have been used in the literature, including e.g. the Bickley Jet. That would save them a lot of challenges in defining the 'exact' solution

6) I am not sure if comparing ends points is the most appropriate metric. Why not compare the along-trajectory differences, which is commonly used in the field (e.g. https://doi.org/10.1029/2018JC014813), so that the full trajectories are taken into account

7) The authors spend a large amount of attention on their 'special-purpose integrators' (section 3.3). However, they don't mention that most implementations of Lagrangian integrators would quite naturally implement such special-purpose integrators, simply because they don't store all time slices in memory so that they need to stop integration on the time of each time slice in order to load the next one

And then also as minor comments

- line 4: clarify here that this is interpolation in space and time?

- line 16: 'computations on data from atmospheric models'?

- line 30: 'For all these applications'?

- line 36: 'capable' is an anthropomorphism; hyperbolic points are not capable of anything.

- line 36: By whom is this recommended?

- lines 43-54 and 126: It might be very useful to include a table with details of the different integrators, so that readers don't need to dig into the literature themselves to find out what the specifics are of each of these

- line 55: 'very common' is perhaps too strong? Lagrangian oceanography is still a bit of a niche

- line 93: why is $x$ not bold here?

- line 104 and other places: Would errors $e$ and $E$ not always be absolute values?

- line 172: At least summarise where these values 2.5 and 0.8 come from

- line 176: How does this extent to staggered grids (e.g. Arakawa-C) which are often used in oceanography?

- line 191: The word 'quite' is somewhat vague here

- line 224: give some examples relevant to Lagrangian oceanography of these cases?

- Figure 1: Also show (some of) the trajectories here, to give readers a feeling for the extent of dispersion?

- line 320: it is unclear at what depth the particles are released, and also whether they are advected in 2D or in 3D

- line 329: is 'transport' the best word here?

- line 363: explain what kind of padding is done

- line 373: explain why this creates additional discontinuities

- line 381: what is the standard deviation/variability around this median? Are the differences between the runs larger than the variability within the runs?

- line 388, 389 and 390: even though the authors make a good point about comparing number of computations instead of runtime, here they still mention that runs are 'longer' and report runtime in seconds.

- Figure 4: I'm a bit confused by the number of points on each of these lines. Why do some have 11 points, even though according to table 1 there were only 9 time steps and 10 tolerances tested?

- Table 2: Would this data be easier to parse in a figure instead of a table?

- line 493: mention what is 'special' about these special-purpose integrators (e.g. 'that don't step across time grids' or something like that)
* * *

---

## Referee Comment (RC2) · Anonymous Referee #2 · 20 Jul 2020

In this paper the authors investigate the accuracy of different ODE integrators for computing drifter trajectories from pre-computed velocity fields, as often used in Lagrangian oceanography. The authors give a detailed and comprehensive background of commonly used methods with fixed timestep sizes, as well as variable timestep methods, and propose a set of special-purpose variations of the latter to better deal with sampling errors due to the temporal discretisation of the flow field. These are of particular interest, since they can increase the accuracy of the method significantly, while reducing the number of "rejected" steps when approaching a temporal discontinuity in the sampled velocity field.

Overall, I very much enjoyed this paper, both, for its detailed background into an important aspect of Lagrangian oceanography, as well as its analysis of the benefits of

the new variations of variable timestep integrators proposed. As the topic discussed is deeply rooted in practical considerations of dealing with pre-computed velocity fields, I find this work highly informative and appropriate for this journal. Nevertheless, I have a few comments that I hope can help to further clarify and improve this paper. Once addressed I would like to recommend this paper for publication.

Comments:

1) Section 4.1: All three data sets have the same constant temporal resolution of 1 hour. Since of of the key advantages of the proposed time-varying integrators is to better treat temporal discontinuities in the sampled flow field, I'm wondering if varying temporal resolutions might have an impact on the analysis as well? Some clarification, either in section 4 or 5 would help here, or possibly even an additional test case with a known analytical solution and different temporal resolutions could be used to high-light this (something akin to 3.2, but comparing fixed / time-varying / special-purpose integrators).

2) Section 4.4: "We used only the surface layer of the data sets", but then 3D spline interpolators are used. Are the experiments considering 2-dimensional trajectories or 3-dimensional ones? Please clarify.

3) Section 5.: "Number of evaluations of the right-hand side was chosen as a measure of work, as it is more objective than the runtime of the simulation" is almost immediately followed by "We note that higher-order interpolation is more computationally costly than lower order interpolation." While both statements are correct in their own context, they seem a little contradictory here. A small clarification could help clarify this.

Moreover, while I agree that the number of evaluations is an important metric to evaluate the efficiency of different numerical integrators, the overall time-to-solution is often the final metric in practice. The final paragraph of 5.2 hints at this, but I'm left wondering if a graph plotting error vs. run-time could be used to highlight the points here more clearly?

[Figure]

4) Section 6: "The most striking conclusion from the results presented above," This reads more like a continuation of the discussion above, rather than a conclusion in its own right. Maybe re-structure a little to independently re-state the objective and key findings of the paper, as is to some extend done later in the section?

Minor details:

* Link to data sets strictcly requires 'https://' in the URL. Please adjust footnote on p. 12.

* The code repository on github is very neat (much appreciated!), but I could not find the Jupyter notebooks mentioned in the text. (In case I just missed them, maybe a link in the README in the repo would help people find them quickly?)

---

## Author Comment (AC1) · 23 Jul 2020

Dear Anonymous Reviewer #1,

Thank you for your comments. We are glad you agree with the motivation for the manuscript, and that you say it would be a welcome addition to the field.

Regarding your comments, we address these below, and we also outline the changes we suggest to make in the manuscript.

[Figure]

**Major comments**

1) The authors make the case for higher-order spatial interpolation, as if that is always better. However, there is no discussion at all about how the order of spatial interpolation is related to the order of the advection schemes within the OGCM from which the data is derived. Intuitively, I'd say that the most appropriate interpolation would be the scheme that most closely mimics the model advection scheme. I'd suggest the authors discuss this.

There are two points in this comment that we would like to address.

First, it is not our intention to advocate higher-order interpolation as always better. There are advantages and disadvantages to all types of interpolation. Among the disadvantages of linear interpolation are the obviously unphysical discontinuous derivatives. Among the disadvantages of higher-degree splines is the increased computational effort per evaluation, and the possibility of overshoot/oscillations. The question we seek to address in this paper is of a numerical nature: "Given an interpolation scheme, which integrator gives the best balance between accuracy and performance?". We touch upon this in lines 333-335, but we agree that this can be made more clear in the manuscript. Towards the end of the introduction, around line 65 in the original manuscript, we will add the following (or something very similar):

"The purpose of our investigation is not to investigate how well different model resolutions and different interpolation schemes reproduce physical drifter trajectories. Rather, we address the purely numerical question of which combinations of integrator and interpolator give the best work–precision balance, for a given resolution."

Second, we feel that the point about using an advection scheme that mimics the ocean model is outside the scope of the paper. Our motivation is to provide guidance towards solving the common problem of integrating trajectories from offline velocity fields. Being a common oceanographic task, diverse ocean models are

used, and each ocean model often has different advection schemes to choose from when configuring the model. For example, a popular ocean model, ROMS, offers four horizontal advection schemes for the user to choose from: second-order centered, fourth-order centered, fourth-order Akima and third-order upwind; e.g. see https://www.myroms.org/wiki/Numerical_Solution_Technique#Horizontal_and_Vertical_Advection.

Advection schemes in ocean modelling is also a topic of active research, and new methods are expected to be introduced. See for example

https://www.sciencedirect.com/science/article/abs/pii/S146350031000106X
https://www.sciencedirect.com/science/article/abs/pii/S1463500311001831
https://www.sciencedirect.com/science/article/abs/pii/S1463500308001510
https://archimer.ifremer.fr/doc/00435/54690/

Additionally, information about the advection scheme is not always readily available for public ocean current data sets. It is therefore unpractical to try to mimic an advection scheme for the general problem of interest studied here. Our intent is to help oceanographers implement an efficient method to integrate any ocean model velocity provided on a rectangular grid, without further concern.

> 2) The authors also but then quickly step over the problem of interpolation near land. Higher-order interpolation would mean that the halo of land is further extended into the ocean (as very briefly mentioned in line 362). I feel that this should be given more discussion. How would this problem compare to the error that the 'consistency of order p of the numerical method is no longer satisfied when the derivatives are not continuous' (lines 222-223)

It is worth commenting on this issue, but a full discussion would, we feel, be outside the scope of the current study. The global integration error is a universal issue that occur in all Lagrangian models using numerical integration of ODEs, while the handling of land is very much application dependent. For applications like LCS calculations, and

global or large-scale transport simulations, interaction with the coastline may be almost negligible, whereas for applications like near-shore oil spills coastline interaction may be very important, and is implemented differently in different oil spill models (see, e.g., https://www.mdpi.com/2077-1312/6/3/104). Hence, it is impossible to say something generally applicable about how the error due to the handling of land cells compares to the global interpolation error.

We will add the following at line 364:

"Note that with higher-degree interpolation schemes, the fact that we set the currents to zero in land cells will have an effect on one or more of the closest cells to the coastline. For applications such as oil spill modelling, where shoreline interactions are important, this may be a disadvantage."

> 3) There is no discussion at all about the widely-used Analytical advection scheme of e.g. https://doi.org/10.5194/gmd-10-1733-2017. How does that scheme compare to the integrators discussed here?

Implementing a Lagrangian particle tracker with an interpolation scheme and an ODE solver can be accomplished very quickly, with just a few tens of lines of code in, e.g., Python or Matlab. As evidenced by the literature, many authors implement their own Lagrangian particle tracking codes in this way, and use different combinations of interpolation and integration schemes. Hence, we feel that the current manuscript provides useful information to the community, even without a detailed discussion/comparison to more advanced schemes.

While we are not very familiar with the advection scheme used in TRACMASS, it is our understanding that it (bi- or tri-) linearly interpolates the current internally in each cell based on vector components at the cell faces (?), and then solves analytically to find the passage of a particle through the interpolated field inside a cell. Rather than using

a specified timestep or tolerance, this approach is more "event driven", calculating the trajectory through a cell as one step.

In the case of trilinear interpolation, it should be possible to compare our trajectories to those obtained with TRACMASS, but for the higher-degree interpolation schemes a direct comparison would be (numerically) meaningless. In any case, we feel that a detailed discussion/comparison to the TRACMASS trajectory scheme would be outside the scope of the current investigation, but might be an interesting topic for a future study.

> 4) The authors mention that they ignore diffusion (line 74). However, in most applications diffusion will be included in the computations. I wonder whether the errors caused by the finite-sized set in the Wiener process are not much larger than any errors in interpolation as discussed here. It would be good if the authors could comment on this.

This is not entirely straightforward. Strictly speaking, advection-diffusion problems are not modelled with ODE methods, but with SDE methods, which are usually of low order. While it might be common in practice to use a higher-order ODE method, and tack on a random displacement in an ad hoc manner, such a splitting of the problem is in itself an approximation which introduces an error. For spatially varying diffusivity, the short timestep required for the SDE method to give a sufficiently small error (in the weak sense) may also render the advection error irrelevant. A thorough discussion of this issue is definitely outside the scope of the paper.

However, it is of course clear that adding random increments to the position of a particle may in many cases dominate the numerical integration errors. In these cases, it would probably give little or no extra benefit to use higher-order integration schemes. We propose to add the following paragraph towards the end of the Discussion (probably in Section 5.3):

[Figure]

"As mentioned in Section 2, we have considered pure advection, ignoring diffusion. However, for many applications diffusion must be included. Solving the advection-diffusion equation with a particle method amounts to numerical solution of a stochastic differential equation (SDE), instead of an ODE. A range of different SDE schemes exist, and the details differ, but all such schemes involve adding a random increment at each timestep. If the random increment is far larger than the local numerical error in each step, then the numerical error in the advection is probably of limited practical importance. The details will depend on the application, and we encourage experimentation. A detailed description of numerical SDE schemes is outside the scope of this study, but the interested reader may find it useful to refer to, e.g., Kloeden Platen (2013), Gräwe et al. (2012) and Spivakovskaya et al. (2007)."

References:

Gräwe, U., Deleersnijder, E., Shah, S. H. A. M., Heemink, A. W. "Why the Euler scheme in particle tracking is not enough: the shallow-sea pycnocline test case." Ocean Dynamics, 62(4) (2012) pp. 501-514.

Kloeden, Peter E., and Eckhard Platen. Numerical solution of stochastic differential equations. Vol. 23. Springer Science Business Media, 2013.

Spivakovskaya, Darya, Arnold W. Heemink, and Eric Deleersnijder. "Lagrangian modelling of multi-dimensional advection-diffusion with space-varying diffusivities: theory and idealized test cases." Ocean Dynamics 57(3) (2007) pp. 189-203.

> 5) I wonder why the authors don't test their method on (complex) flows where an exact solution is known. Quite a few of these flows have been used in the literature, including e.g. the Bickley Jet. That would save them a lot of challenges in defining the 'exact' solution.

An important motivation for avoiding an analytical reference solution, is that it seems

likely that fifth-degree spline interpolation would out-perform cubic splines and linear interpolation in terms of accuracy, when comparing to an analytical solution. That would lead to the conclusion that higher-degree splines give more accurate results, which is by no means certain for ocean currents. By using numerically obtained reference solutions, obtained separately for each interpolation scheme, we remain "interpolation-agnostic", merely addressing the question of which integrator/interpolator pair is numerically more efficient.

As mentioned in our response to comment 1, our intent is to help oceanographers implement an efficient method to integrate any ocean model velocity. Hence, we wanted the discussion and conclusions to feel directly relevant to applied oceanographers. For example, we conclude that the most efficient numerical approach for a given level of accuracy depends on the spatial resolution of the dataset. If we were to start with an analytically defined flowfield, and then discretely evaluate this on different grids, for later interpolation, it is not obvious how any conclusions could be applied directly to ocean current datasets.

> 6) I am not sure if comparing ends points is the most appropriate metric. Why not compare the along-trajectory differences, which is commonly used in the field (e.g. https://doi.org/10.1029/2018JC014813), so that the full trajectories are taken into account.

The endpoints were chosen simply because of the discussion in terms of the theory for numerical solution of ODEs. The standard approach in the ODE literature is to discuss convergence in terms of the global error, the order of a method refers to the global error, etc. In Figure 4, the error as a function of number of evaluations for the fixed-step integrators make straight lines in the log-log plot, with a slope determined by the order of convergence. If we considered the along-trajectory error, there would be less of a direct link to the theory of ODE methods when discussing these results.

From an application point of view, different metrics could be appropriate. For, e.g, LCS applications, only the endpoint matters, while for other applications the entire particle history might be relevant. In light of these points, we feel that the end points are, on the whole, the most appropriate metric.

> 7) The authors spend a large amount of attention on their 'special-purpose integrators' (section 3.3). However, they don't mention that most implementations of Lagrangian integrators would quite naturally implement such special-purpose integrators, simply because they don't store all time slices in memory so that they need to stop integration on the time of each time slice in order to load the next one.

We believe this is a misunderstanding. Taking the example of cubic interpolation, one needs at least 4 time slices in memory to construct the interpolation. In the case of our special-purpose integrators, integration is stopped and restarted at every one of those time slices, not just when new data must be loaded.

For linear interpolation, two time slices will suffice to construct the interpolator, but in for example Taylor and Shadden (2008), which uses linear basis function interpolation and a variable-step Runge-Kutta-Fehlberg method, there is no description of stopping and restarting integration at every time slice. In our opinion, it is often hard to be sure of the exact details of how the full trajectory calculation has been implemented, given the typically very short descriptions of interpolation and integration schemes in the applied literature. Hence, a thorough discussion should be of value to the community, even if the method itself were to have been used by others before.

For fixed-step integrators, we do discuss the fact that these perform very well when the timestep is chosen such that integration is stopped and restarted at every time slice. See, e.g., lines 415–420 in the Discussion, and 515–520 in the Conclusion.

**Minor comments**

- line 4: clarify here that this is interpolation in space and time?

OK.

- line 16: 'computations on data from atmospheric models'?

OK.

- line 30: 'For all these applications'?

OK.

- line 36: 'capable' is an anthropomorphism; hyperbolic points are not capable of anything.

Will rephrase.

- line 36: By whom is this recommended?

Agree that this is unclear, will add reference or rephrase.

- lines 43-54 and 126: It might be very useful to include a table with details of the different integrators, so that readers don't need to dig into the literature themselves to find out what the specifics are of each of these

This comment refers to the introduction, where we mention different integrators used in the applied literature. Given that these include not only Runge-Kutta methods, but also linear multistep methods and predictor-corrector methods, a useful description of these would probably require several pages. We don't see that a table could contain enough information to be useful, and for anyone who wants to implement these methods, looking them up would not be a large amount of work.

> - line 55: 'very common' is perhaps too strong? Lagrangian oceanography is still a bit of a niche

Will drop "very".

> - line 93: why is $x$ not bold here?

We have left $x$ in non-bold for the general theory discussion. Will add a sentence to make this more clear.

> - line 104 and other places: Would errors $e$ and $E$ not always be absolute values?

No. The error it self can be either positive or negative (see, e.g., Eq. (3.1) in Hairer et al. (1993)). However, when we discuss how the error scales with the timestep, it's the absolute value.

> - line 172: At least summarise where these values 2.5 and 0.8 come from

This is a mistake on our part, as the reference given doesn't actually discuss the choice in detail. Will rephrase line 172 as follows:

"The factors 0.8 and 2.5 were chosen from a range of values recommended by Hairer et al. (1993, p. 168), and were kept constant for all numerical experiments."

  - line 176: How does this extent to staggered grids (e.g. Arakawa-C) which are often used in oceanography?

Simply interpolate each vector component on its own grid. The vector components are interpolated separately, so this would require only a minimal change at the initialisation of the interpolators in our implementation.

  - line 191: The word 'quite' is somewhat vague here

OK, will drop the word "quite".

  - line 224: give some examples relevant to Lagrangian oceanography of these cases?

We propose to add the following at the end of line 225:

"The more pathological examples are probably not likely to occur in practice. However, as we will see later, the unbounded error in a single step can in some cases dominate the global error, making the use of a higher-order scheme pointless."

  - Figure 1: Also show (some of) the trajectories here, to give readers a feeling for the extent of dispersion?

We assume this comment is meant to refer to Figure 3? We have created a figure (attached) showing the final positions of the particles, for the three different datasets. We propose to add this figure either at the start of Section 5, or in an appendix.

- line 320: it is unclear at what depth the particles are released, and also whether they are advected in 2D or in 3D

OK. It is stated in line 360 that we use only the surface layer of the currents, but this should be described clearly in Section 4.1 or 4.2 as well. We propose adding the following after line 300:

"We have chosen to consider two-dimensional (horizontal) transport only, using the surface layer of the modelled current data. The current velocity field is interpolated in three dimensions (two spatial dimensions plus time), using the same degree of interpolation in all three dimensions."

- line 329: is 'transport' the best word here?

We propose to change this to "trajectories".

- line 363: explain what kind of padding is done

Padding was probably not the right word. It only means that when cropping the dataset, the subset selected was larger than then minimal size required to cover all the trajectories. Will rephrase.

- line 373: explain why this creates additional discontinuities

This is somewhat technical, and hard to explain without going into details on how spline interpolations are constructed, which we have not covered in the paper. It is also a bit on the side, as we don't investigate this type of interpolator. Therefore, we would prefer not to go into detail, and simply refer the interested reader to Lekien and Marsden.

- line 381: what is the standard deviation/variability around this median? Are the differences between the runs larger than the variability within the runs?

Not sure how to best show the standard deviation. The plots in Fig. 4 would become unreadable if we added error bars to every datapoint. (Also, recall that everything is completely deterministic. Therefore, the "variability within the runs" comes only from the different initial positions.)

An option might be to add this as supplementary information, either in the form of several figures, or a table.

- line 388, 389 and 390: even though the authors make a good point about comparing number of computations instead of runtime, here they still mention that runs are 'longer' and report runtime in seconds.

That is only meant to illustrate that each evaluation takes longer when higher-degree interpolation is used. We will make this more clear.

- Figure 4: I'm a bit confused by the number of points on each of these lines. Why do some have 11 points, even though according to table 1 there were only 9 time steps and 10 tolerances tested?

This is clearly an inconsistency between Table 1 and the figures. We will fix this.

- Table 2: Would this data be easier to parse in a figure instead of a table?

In our opinion, a table is most suited in this case.

- line 493: mention what is 'special' about these special-purpose integrators
(e.g. 'that don't step across time grids' or something like that)

Ok, will expand/rephrase.

[Figure]

**Fig. 1.**

---

## Author Comment (AC2) · 30 Jul 2020

Dear Anonymous Reviewer #2,

Thank you for your encouraging comments. Below, we address the suggested changes to our manuscript.

**Major comments**

1) Section 4.1: All three data sets have the same constant temporal resolution of 1 hour. Since of of the key advantages of the proposed time-varying

integrators is to better treat temporal discontinuities in the sampled flow field, I'm wondering if varying temporal resolutions might have an impact on the analysis as well? Some clarification, either in section 4 or 5 would help here, or possibly even an additional test case with a known analytical solution and different temporal resolutions could be used to highlight this (something akin to 3.2, but comparing fixed / time-varying / special-purpose integrators).

There is indeed an effect of temporal resolution. We did some tests initially, but these were not included in the manuscript. While we would have to double check to be sure, the improvement seen for the special-purpose integrators is probably smaller for datasets with coarser temporal resolution. The reason is the relative importance of crossing discontinuities in the time dimension relative to the spatial dimensions. This is discussed in lines 429-437, when discussing the different spatial resolutions.

It is certainly possible to downsample the datasets to for example 3 hours or 6 hours timesteps, and re-run the simulations. This will make the paper a bit longer, but it might also make the paper more generally relevant. We will need some time to run the simulations and include the results in the paper.

Any further comments on this point by the reviewer are most welcome.

2) Section 4.4: "We used only the surface layer of the data sets", but then 3D spline interpolators are used. Are the experiments considering 2-dimensional trajectories or 3-dimensional ones? Please clarify.

Time is the third dimension used in the interpolator. This is mentioned in lines 368/369, but we will make this more clear earlier in section 4, in line with our reply to comments by Reviewer 1. We will add the following after line 300:

"We have chosen to consider two-dimensional (horizontal) transport only, using the surface layer of the modelled current data. The current velocity field is interpolated in three dimensions (two spatial dimensions plus time), using the same degree of interpolation in all three dimensions."

> 3) Section 5.: "Number of evaluations of the right-hand side was chosen as a measure of work, as it is more objective than the runtime of the simulation" is almost immediately followed by "We note that higher-order interpolation is more computationally costly than lower order interpolation." While both statements are correct in their own context, they seem a little contradictory here. A small clarification could help clarify this. Moreover, while I agree that the number of evaluations is an important metric to evaluate the efficiency of different numerical integrators, the overall time-to-solution is often the final metric in practice. The final paragraph of 5.2 hints at this, but I'm left wondering if a graph plotting error vs. run-time could be used to highlight the points here more clearly?

We will change the second sentence, to read

"We note, however, that higher-order interpolation is more computationally costly to evaluate than lower-order, and thus the same number of evaluations will take longer when using an interpolator of higher degree."

We can also add a version of figure 4 using run-time on the horizontal axis, as an appendix.

> 4) Section 6: "The most striking conclusion from the results presented above," This reads more like a continuation of the discussion above, rather than a conclusion in its own right. Maybe re-structure a little to independently re-state the objective and key findings of the paper, as is to some extend done later in the section?

We will rephrase the start of this section.

**Minor comments**

* Link to data sets strictcly requires 'https://' in the URL. Please adjust foot-note on p.12.

Ok.

* The code repository on github is very neat (much appreciated!), but I could not find the Jupyter notebooks mentioned in the text. (In case I just missed them, maybe a link in the README in the repo would help people find them quickly?)

The notebooks are indeed missing. Our apologies. We will make sure the repo is complete, and create a "release" with its own DOI that can be cited, prior to submitting the final manuscript.

---

## Author Response (AR1)

**Response to reviewers**

Dear Editor, Dear Reviewers,

Thank you for considering our manuscript for publication in Geoscientific Model Development. Below, we address in detail each point raised by the reviewers, and describe the changes we have made to the manuscript in response to those points. In addition, we have made a few other changes, the most important being an added sentence in the abstract, and a reference to our simulation code, of which we have created a permanently available version with a DOI from zenodo. All the changes are visible in the marked-up manuscript provided further down in this pdf.

**Reviewer 1**

Thank you for your comments. We are glad you agree with the motivation for the manuscript, and that you say it would be a welcome addition to the field. Regarding your comments, we address these below, and we also outline the changes we suggest to make in the manuscript.

**Major comments**

**Comment 1**

> The authors make the case for higher-order spatial interpolation, as if that is always better. However, there is no discussion at all about how the order of spatial interpolation is related to the order of the advection schemes within the OGCM from which the data is derived. Intuitively, I'd say that the most appropriate interpolation would be the scheme that most closely mimics the model advection scheme. I'd suggest the authors discuss this.

There are two points in this comment that we would like to address. First, it is not our intention to advocate higher-order interpolation as always better. There are advantages and disadvantages to all types of interpolation. Among the disadvantages of linear interpolation are the obviously unphysical discontinuous derivatives. Among the disadvantages of higher-degree splines is the increased computational effort per evaluation, and the possibility of overshoot/oscillations. The question we seek to address in this paper is of a numerical nature: "Given an interpolation scheme, which integrator gives the best balance between accuracy and performance?". We touch upon this in lines 333-335, but we agree that this can be made more clear in the manuscript. Towards the end of the introduction, around line 65 in the original manuscript, we have added the following:

"We note that the purpose of our investigation is not to determine how well different model resolutions and different interpolation schemes reproduce physical drifter trajectories. Rather, we address the purely numerical question of which combinations of integrator and interpolator give the best work–precision balance, for a given resolution."

Second, we feel that the point about using an advection scheme that mimics the ocean model is outside the scope of the paper. Our motivation is to provide guidance towards solving the common problem of integrating trajectories from offline velocity fields. Being a common oceanographic task, diverse ocean models are used, and each ocean model often has different advection schemes to choose from when configuring the model. For example, a popular ocean model, ROMS, offers four horizontal advection schemes for the user to choose from: second-order centered, fourth-order centered, fourth-order Akima and third-order upwind; e.g. see `https://www.myroms.org/wiki/Numerical_Solution_Technique#Horizontal_and_Vertical_Advection`. Advection schemes in ocean modelling is also a topic of active research, and new methods are expected to be introduced. See for example

https://www.sciencedirect.com/science/article/abs/pii/S146350031000106X
https://www.sciencedirect.com/science/article/abs/pii/S1463500311001831
https://www.sciencedirect.com/science/article/abs/pii/S1463500308001510
https://archimer.ifremer.fr/doc/00435/54690/

Additionally, information about the advection scheme is not always readily available for public ocean current data sets. It is therefore impractical to try to mimic an advection scheme for the general problem of interest studied here. Our intent is to help oceanographers implement an efficient method to integrate any ocean model velocity provided on a rectangular grid, without further concern.

**Comment 2**

> The authors also but then quickly step over the problem of interpolation near land. Higher-order interpolation would mean that the halo of land is further extended into the ocean (as very briefly mentioned in line 362). I feel that this should be given more discussion. How would this problem compare to the error that the 'consistency of order p of the numerical method is no longer satisfied when the derivatives are not continuous' (lines 222-223)

It is worth commenting on this issue, but a full discussion would, we feel, be outside the scope of the current study. The global integration error is a universal issue that occur in all Lagrangian models using numerical integration of ODEs, while the handling of land is very much application dependent. For applications like LCS calculations, and global or large-scale transport simulations, interaction with the coastline may be almost negligible, whereas for applications like near-shore oil spills coastline interaction may be very important, and is implemented differently in different oil spill models (see, e.g., https://www.mdpi.com/2077-1312/6/3/104). Hence, it is impossible to say something generally applicable about how the error due to the handling of land cells compares to the global interpolation error.

We have added the following at line 364 in the original manuscript:

"Note that with higher-degree interpolation schemes, the fact that we set the currents to zero in land cells will have an effect on one or more of the closest cells to the coastline. For applications such as oil spill modelling, where shoreline interactions are important, a different strategy might be needed."

**Comment 3**

> There is no discussion at all about the widely-used Analytical advection scheme of e.g. `https://doi.org/10.5194/gmd-10-1733-2017`. How does that scheme compare to the integrators discussed here?

Implementing a Lagrangian particle tracker with an interpolation scheme and an ODE solver can be accomplished very quickly, with just a few tens of lines of code in, e.g., Python or Matlab. As evidenced by the literature, many authors implement their own Lagrangian particle tracking codes in this way, and use different combinations of interpolation and integration schemes. Hence, we feel that the current manuscript provides useful information to the community, even without a detailed discussion/comparison to more advanced schemes.

While we are not very familiar with the advection scheme used in TRACMASS, it is our understanding that it (bi- or tri-) linearly interpolates the current internally in each cell based on vector components at the cell faces, and then solves analytically to find the passage of a particle through the interpolated field inside a cell. Rather than using a specified timestep or tolerance, this approach analytically calculates the trajectory through a cell as one step.

In the case of trilinear interpolation, it should be possible to compare our trajectories to those obtained with TRACMASS, but for the higher-degree interpolation schemes a direct comparison would be (numerically) meaningless. In any case, we feel that a detailed discussion/comparison to the TRACMASS trajectory scheme would be outside the scope of the current investigation, but might be an interesting topic for a future study.

**Comment 4**

> The authors mention that they ignore diffusion (line 74). However, in most applications diffusion will be included in the computations. I wonder whether the errors caused by the

finite-sized set in the Wiener process are not much larger than any errors in interpolation as discussed here. It would be good if the authors could comment on this.

There are two points we would like to raise, in response to this comment. First a point on applications, and second an numerical/theoretical point.

First, although it is true that the addition of diffusion is common in many applications, there are also many oceanographic and atmospheric studies which will want to compute trajectories without diffusion. One example includes all LCS computations, as a requirement for LCS theory to hold is that the velocity be deterministic.

Furthermore, there are studies that deal with Lagrangian trajectories that intentionally do not include diffusivity. Examples include using backwards trajectories to investigate the source of particles that end up in the sediments in a particular location, and using Lagrangian trajectories to study the path and history of water arriving at a particular location of upwelling:

`www.nature.com/articles/ncomms7521`
`journals.ametsoc.org/jpo/article/41/1/88/11315/A-Numerical-Modeling-Study-of-the-Upwelling-Source`

Second, the issue of adding diffusion is not entirely straightforward theoretically and numerically. Strictly speaking, advection-diffusion problems are not modelled with ODE methods, but with SDE methods, which are usually of lower order. While it might be common in practice to use a higher-order ODE method, and tack on a random displacement in an *ad hoc* manner, such a splitting of the problem is in itself an approximation which introduces an error. For spatially varying diffusivity, the short timestep required for the SDE method to give a sufficiently small error (in the weak sense) may also render the advection error irrelevant. A thorough discussion of this issue is definitely outside the scope of the paper.

However, it is of course clear that adding random increments to the position of a particle may in many cases dominate the numerical integration errors. In these cases, it would probably give little or no extra benefit to use higher-order integration schemes. To address this issue a bit more, we have added a subsection in the discussion, with the following two paragraphs (Section 5.3 in the revised manuscript):

"As mentioned in Section 2, we have considered pure advection, ignoring diffusion. Calculating trajectories with pure advection by a deterministic velocity field is common in several applications, perhaps most notably for identification of LCS (see, e.g., Haller (2015), Allshouse et al. (2017), Duran et al. (2018)). Other examples include the use of backwards trajectories to identify source regions for particles ending up in the sediments (van Sebille et al., (2015), and analysis of Lagrangian pathways to study the source and history of water parcels reaching a particular upwelling zone (Rivas & Samelson, 2011). In general, simulating diffusion in Lagrangian oceanography (or meteorology) may introduce a complication that encourages some studies to compute trajectories without diffusion: Lagrangian motion becomes ambiguous when diffusive mixing is simulated, because the identity of a fluid parcel is lost. On the other hand, ignoring small-scale mixing may also be problematic. One approach to this problem is to supplement purely advective trajectories with along-path changes in parcel properties, as discussed in Rivas & Samelson (2011).

However, for many other applications diffusion must be included. Solving the advection-diffusion equation with a particle method amounts to numerical solution of a stochastic differential equation (SDE), instead of an ODE. A range of different SDE schemes exist, and the details differ, but all such schemes involve adding a random increment at each timestep. If the random increment is far larger than the local numerical error in each step, then the numerical error in the advection is probably of limited practical importance. The details will depend on the application, and we encourage experimentation. A detailed description of numerical SDE schemes is outside the scope of this study, but the interested reader may find it useful to refer to, e.g., Kloeden & Platen (1992), Spivakovskaya et al. (2007), and Gräwe (2011)."

**Comment 5**

> I wonder why the authors don't test their method on (complex) flows where an exact solution is known. Quite a few of these flows have been used in the literature, including e.g. the Bickley Jet. That would save them a lot of challenges in defining the 'exact' solution.

An important motivation for avoiding an analytical reference solution, is that it seems likely that fifth-degree spline interpolation would out-perform cubic splines and linear interpolation in terms of accuracy, when comparing to an analytical solution. That would lead to the conclusion that higher-degree splines

give more accurate results, which is by no means certain for ocean currents. By using numerically obtained reference solutions, obtained separately for each interpolation scheme, we remain "interpolation-agnostic", merely addressing the question of which integrator/interpolator pair is numerically more efficient.

As mentioned in our response to comment 1, our intent is to help oceanographers implement an efficient method to integrate any ocean model velocity. Hence, we wanted the discussion and conclusions to feel directly relevant to applied oceanographers. For example, we conclude that the most efficient numerical approach for a given level of accuracy depends on the spatial resolution of the dataset. If we were to start with an analytically defined flowfield, and then discretely evaluate this on different grids, for later interpolation, it is not obvious how any conclusions could be applied directly to ocean current datasets.

**Comment 6**

I am not sure if comparing ends points is the most appropriate metric. Why not compare the along-trajectory differences, which is commonly used in the field (e.g. `https://doi.org/10.1029/2018JC014813`), so that the full trajectories are taken into account.

The endpoints were chosen simply because of the discussion in terms of the theory for numerical solution of ODEs. The standard approach in the ODE literature is to discuss convergence in terms of the global error, the order of a method refers to the global error, etc. In Figure 4, the error as a function of number of evaluations for the fixed-step integrators make straight lines in the log-log plot, with a slope determined by the order of convergence. If we considered the along-trajectory error, there would be less of a direct link to the theory of ODE methods when discussing these results.

From an application point of view, different metrics could be appropriate. For, e.g, LCS applications, only the endpoint matters, while for other applications the entire particle history might be relevant. In light of these points, we feel that the end points are, on the whole, the most appropriate metric.

**Comment 7**

The authors spend a large amount of attention on their 'special-purpose integrators' (section 3.3). However, they don't mention that most implementations of Lagrangian integrators would quite naturally implement such special-purpose integrators, simply because they don't store all time slices in memory so that they need to stop integration on the time of each time slice in order to load the next one.

We believe this is a misunderstanding by the reviewer. Taking the example of cubic interpolation, one needs at least 4 time slices in memory simultaneously to construct the interpolation. In the case of our special-purpose integrators, integration is stopped and restarted at every one of those time slices, not just when new data must be loaded.

For linear interpolation, two time slices will suffice to construct the interpolator, but in for example Taylor and Shadden (2008), which uses linear basis function interpolation and a variable-step Runge-Kutta-Fehlberg method, there is no description of stopping and restarting integration at every time slice. In our opinion, it is often hard to be sure of the exact details of how the full trajectory calculation has been implemented, given the typically very short descriptions of interpolation and integration schemes in the applied literature. Hence, a thorough discussion should be of value to the community, even if the method itself were to have been used by others before.

For fixed-step integrators, we do discuss the fact that these perform very well when the timestep is chosen such that integration is stopped and restarted at every time slice. See, e.g., lines 415–420 in the Discussion, and 515–520 in the Conclusion.

**Minor comments**

- line 4: clarify here that this is interpolation in space and time?

OK.

- line 16: 'computations on data from atmospheric models'?

OK.

- line 30: 'For all these applications'?

OK.

- line 36: 'capable' is an anthropomorphism; hyperbolic points are not capable of anything.

We have rephrased this to "... presenting hyperbolic points where initially small errors may grow exponentially.".

- line 36: By whom is this recommended?

We have rephrased to "It may therefore be useful to employ higher-order integration methods ...".

- lines 43-54 and 126: It might be very useful to include a table with details of the different integrators, so that readers don't need to dig into the literature themselves to find out what the specifics are of each of these

This comment refers to the introduction, where we mention different integrators used in the applied literature. Given that these include not only Runge-Kutta methods, but also linear multistep methods and predictor-corrector methods, a useful description of these would probably require several pages. We don't see that a table could contain enough information to be useful, and for anyone who wants to implement these methods, looking them up would not be a large amount of work.

- line 55: 'very common' is perhaps too strong? Lagrangian oceanography is still a bit of a niche

We have removed "very".

- line 93: why is $x$ not bold here?

We have left $x$ in non-bold for the general theory discussion. We have added a brief explanation to make this more clear:

"In the following, we introduce some elements from the theory of numerical integration of ODEs, which will be needed for the later discussion. While elsewhere in this paper, we consider $\mathbf{x}(t)$ as a two-dimensional vector giving the position of a particle in a horizontal plane, we here simply use $x(t)$, as the theory is general and can be applied to vectors and scalars alike.".

- line 104 and other places: Would errors $e$ and $E$ not always be absolute values?

No. The error it self can be either positive or negative (see, e.g., Eq. (3.1) in Hairer et al. (1993)). However, when we discuss how the error scales with the timestep, it's the absolute value.

- line 172: At least summarise where these values 2.5 and 0.8 come from

Here there are two mistakes on our part, as the reference given doesn't actually discuss the choice in detail, and the parameters used in the code were actually 0.9 and 3.0. We have rephrased line 172 as follows:

"The factors 0.9 and 3.0 were chosen from a range of values recommended by Hairer et al. (1993, p. 168), and were kept constant for all numerical experiments."

- line 176: How does this extent to staggered grids (e.g. Arakawa-C) which are often used in oceanography?

Simply interpolate each vector component on its own grid. The vector components are interpolated separately, so this would require only a minimal change at the initialisation of the interpolators in our implementation.

- line 191: The word 'quite' is somewhat vague here

OK, will drop the word "quite".

- line 224: give some examples relevant to Lagrangian oceanography of these cases?

We have added the following at the end of line 225:

"The more pathological examples are perhaps unlikely to occur in practice. However, as we will see later, when the error in even a single step is unbounded by Eq. (7), this can is some cases dominate the global error, rendering the use of a higher-order scheme pointless."

- Figure 1: Also show (some of) the trajectories here, to give readers a feeling for the extent of dispersion?

We assume this comment is meant to refer to Figure 3? We have created a figure showing the final positions of the particles, for the three different datasets, which has been added as Fig. 4 in the revised manuscript.

- line 320: it is unclear at what depth the particles are released, and also whether they are advected in 2D or in 3D

OK. It is stated in line 360 of the original manuscript that we use only the surface layer of the currents, but we agree that this should be clearly described early in Section 4 as well. We have added the following at the end of the first paragraph of Section 4:

"We have chosen to consider two-dimensional (horizontal) transport only, using the surface layer of the modelled current data. The current velocity field is interpolated in three dimensions (two spatial dimensions plus time), using the same degree of interpolation in all three dimensions."

- line 329: is 'transport' the best word here?

We have changed this to "trajectories".

- line 363: explain what kind of padding is done

Padding was probably not the right word. It only means that when cropping the dataset, the subset selected was larger than then minimal size required to cover all the trajectories. We have rephrased this.

- line 373: explain why this creates additional discontinuities

This is somewhat technical, and hard to explain without going into details on how spline interpolations are constructed, which we have not covered in the paper. It is also a bit on the side, as we don't investigate this type of interpolator. Therefore, we would prefer not to go into detail, and simply refer the interested reader to Lekien and Marsden.

- line 381: what is the standard deviation/variability around this median? Are the differences between the runs larger than the variability within the runs?

We have created a new figure, added as Fig. B1 in the revised manuscript, where we show the range covering 90% of the errors. As this makes the figure somewhat more cluttered, we feel it is best to keep the original figure as it is, and add this new figure as a full-page figure in the Appendix. A reference to this new figure has been added in the first paragraph of Section 5.

- line 388, 389 and 390: even though the authors make a good point about comparing number of computations instead of runtime, here they still mention that runs are 'longer' and report runtime in seconds.

That is only meant to illustrate that each evaluation takes longer when higher-degree interpolation is used. However, in response to a comment by Reviewer 2, we have added an additional figure in the appendix showing error as a function of runtime, and we have expanded and rephrased the second paragraph of Section 5 as follows:

"Number of evaluations of the right-hand side was chosen as a measure of work, as it is more objective than the runtime of the simulation, which would depend on the particular machine used to run the simulations, and also be more susceptible to somewhat random variations. However, for the interested reader we show the error as a function of runtime in Fig. B2.

While we analyse the results in terms of number of function calls, we note that higher-order interpolation is more computationally costly than lower-order interpolation. This means that the same number of evaluations will take more time if a higher degree of interpolation is used. We found that for the simulations done with the fixed-step 4th-order Runge-Kutta integrator, the simulations with cubic spline interpolation took on average four to five times longer than those with linear interpolation, and the simulations with quintic spline interpolation took on average three to four times longer than those with cubic spline interpolation."

- Figure 4: I'm a bit confused by the number of points on each of these lines. Why do some have 11 points, even though according to table 1 there were only 9 time steps and 10 tolerances tested?

This was an inconsistency between Table 1 and the figures, where we had included a tolerance of $10^{-14}$ in the results, but forgotten to add it to Table 1. We have fixed this.

> - Table 2: Would this data be easier to parse in a figure instead of a table?

In our opinion, a table is most suited in this case.

> - line 493: mention what is 'special' about these special-purpose integrators (e.g. 'that don't step across time grids' or something like that)

We have rephrased the end of the second paragraph is Section 6 as follows:

"... This is achieved by stopping and restarting the integration exactly at the grid points of the dataset along the time dimension. By doing this, we avoid stepping across discontinuities in the (higher) derivatives of the velocity field, and we thus avoid picking up local errors that are unbounded by Eq. (7) at those points.".

**Reviewer 2**

Thank you for your encouraging comments. Below, we address the suggested changes to our manuscript.

**Major comments**

**Comment 1**

> Section 4.1: All three data sets have the same constant temporal resolution of 1 hour. Since of of the key advantages of the proposed time-varying integrators is to better treat temporal discontinuities in the sampled flow field, I'm wondering if varying temporal resolutions might have an impact on the analysis as well? Some clarification, either in section 4 or 5 would help here, or possibly even an additional test case with a known analytical solution and different temporal resolutions could be used to highlight this (something akin to 3.2, but comparing fixed / time-varying / special-purpose integrators).

There is some effect of temporal resolution. We did some tests initially, but these were not completed or included in the manuscript, as we felt it would be too long. We also re-ran some of those tests now (with the same datasets, but downsampled to 6 hours temporal resolution), and from a quick analysis, the results appear relatively similar to those presented here. The improvement seen for the special-purpose integrators appears a little smaller for datasets with coarser temporal resolution, but not by a large amount. The reason is the relative importance of crossing discontinuities in the time dimension relative to the spatial dimensions. This is discussed in lines 429-437 of the original manuscript, when discussing the different spatial resolutions.

While the effect of temporal resolution would be an interesting investigation for a future study, we have concluded that we would rather not include it in the current paper, mainly for reasons of length. Presenting the new results takes some space, new reference solutions must be established and documented, and the discussion would have to be expanded. We would also run into questions about resolving the tides, and in particular when the dataset with 800 m resolution is downsampled to 6 hours temporal resolution, the typical current velocities of about 0.25 m/s will move many cells during a timestep, meaning that the low temporal resolution might be inadequate to capture the dynamics of the high-resolution data. All in all, we feel that a thorough discussion of these issues would make the paper too bloated.

**Comment 2**

> Section 4.4: "We used only the surface layer of the data sets", but then 3D spline interpolators are used. Are the experiments considering 2-dimensional trajectories or 3-dimensional ones? Please clarify.

Time is the third dimension used in the interpolator. This is mentioned in lines 368/369, but we will make this more clear earlier in section 4, in line with our reply to comments by Reviewer 1. We have added the following at the end of the first paragraph of Section 4:

"We have chosen to consider two-dimensional (horizontal) transport only, using the surface layer of the modelled current data. The current velocity field is interpolated in three dimensions (two spatial dimensions plus time), using the same degree of interpolation in all three dimensions."

**Comment 3**

> Section 5.: "Number of evaluations of the right-hand side was chosen as a measure of work, as it is more objective than the runtime of the simulation" is almost immediately followed by "We note that higher-order interpolation is more computationally costly than lower order interpolation." While both statements are correct in their own context, they seem a little contradictory here. A small clarification could help clarify this. Moreover, while I agree that the number of evaluations is an important metric to evaluate the efficiency of different numerical integrators, the overall time-to-solution is often the final metric in practice. The final paragraph of 5.2 hints at this, but I'm left wondering if a graph plotting error vs. run-time could be used to highlight the points here more clearly?

We have added an additional figure in the appendix showing error as a function of runtime, and we have expanded and rephrased the second paragraph of Section 5 as follows:

"Number of evaluations of the right-hand side was chosen as a measure of work, as it is more objective than the runtime of the simulation, which would depend on the particular machine used to run the simulations, and also be more susceptible to somewhat random variations. However, for the interested reader we show the error as a function of runtime in Fig. B2.

While we analyse the results in terms of number of function calls, we note that higher-order interpolation is more computationally costly than lower-order interpolation. This means that the same number of evaluations will take more time if a higher degree of interpolation is used. We found that for the simulations done with the fixed-step 4th-order Runge-Kutta integrator, the simulations with cubic spline interpolation took on average four to five times longer than those with linear interpolation, and the simulations with quintic spline interpolation took on average three to four times longer than those with cubic spline interpolation."

**Comment 4**

> Section 6: "The most striking conclusion from the results presented above," This reads more like a continuation of the discussion above, rather than a conclusion in its own right. Maybe re-structure a little to independently re-state the objective and key findings of the paper, as is to some extend done later in the section?

We have expanded and rephrased the start of Section 6 as follows:

"In this paper, we have investigated how different numerical integrators behave, in combination with different degrees of interpolation, and datasets of different spatial resolution. We have calculated trajectories over 72 hours, from 10000 initial positions, and compared the integrator-interpolator pairs in terms of the error in the final position of each trajectory. We have considered linear, cubic and quintic spline interpolation, along with four fixed-step Runge-Kutta integrators of orders 1 to 4, three commonly used variable-step integrators, and three special-purpose variants of the latter.

The most striking conclusion from our results is that the special-purpose integrators we describe in many cases deliver several orders of magnitude more accurate results, at no additional cost. Alternatively, they can deliver the same accuracy as standard methods, with highly reduced computational effort. This is achieved by stopping and restarting the integration exactly at the grid points of the dataset along the time dimension. By doing this, we avoid stepping across discontinuities in the (higher) derivatives of the velocity field, and thus we avoid picking up local errors that are unbounded by Eq. (7) at those points."

**Minor comments**

> * Link to data sets strictcly requires 'https://' in the URL. Please adjust footnote on p.12.

This has been fixed.

* The code repository on github is very neat (much appreciated!), but I could not find the Jupyter notebooks mentioned in the text. (In case I just missed them, maybe a link in the README in the repo would help people find them quickly?)

The notebooks were indeed missing. Our apologies. We have added the notebooks to the repo, and created a "release" with a DOI that can be cited.

[revised manuscript text omitted]